# Rad51-mediated replication of damaged templates relies on monoSUMOylated DDK kinase

Chinnu Rose Joseph[1], Sabrina Dusi [1], Michele Giannattasio [1,2] & Dana Branzei [1,3 ✉]

DNA damage tolerance (DDT), activated by replication stress during genome replication, is mediated by translesion synthesis and homologous recombination (HR). Here we uncover that DDK kinase, essential for replication initiation, is critical for replication-associated recombination-mediated DDT. DDK relies on its multi-monoSUMOylation to facilitate HR-mediated DDT and optimal retention of Rad51 recombinase at replication damage sites. Impairment of DDK kinase activity, reduced monoSUMOylation and mutations in the putative SUMO Interacting Motifs (SIMs) of Rad51 impair replication-associated recombination and cause fork uncoupling with accumulation of large single-stranded DNA regions at fork branching points. Notably, genetic activation of salvage recombination rescues the uncoupled fork phenotype but not the recombination-dependent gap-filling defect of DDK mutants, revealing that the salvage recombination pathway operates preferentially proximal to fork junctions at stalled replication forks. Overall, we uncover that monoSUMOylated DDK acts with Rad51 in an axis that prevents replication fork uncoupling and mediates recombination-dependent gap-filling.

[1] IFOM, Istituto Fondazione di Oncologia Molecolare, Via Adamello 16, 20139 Milan, Italy. [2] Università degli Studi di Milano, Dipartimento di Oncologia ed Emato-Oncologia, Via S. Sofia 9/1, 20122 Milano, Italy. [3] Istituto di Genetica Molecolare, Consiglio Nazionale delle Ricerche (IGM-CNR), 27100 Pavia, Italy. ✉email: dana.branzei@ifom.eu

A high number of DNA lesions occur spontaneously daily in every single cell[1]. Most of these lesions are efficiently removed by excision repair pathways[2,3], but some DNA lesions persist and interfere with the functioning of DNA polymerases during chromosome replication[4]. Various sources of replication stress and exogenous types of DNA damaging agents, such as UV irradiation and chemotherapeutic agents, increase the load of DNA damage experienced by cells during chromosome replication. To get around these different types of lesions encountered during DNA replication, cells utilize DNA damage tolerance (DDT) mechanisms, divided broadly into trans-lesion synthesis (TLS) and homologous recombination (HR)-mediated bypass by template switching[5]. TLS is mediated by mutagenic error-prone polymerases, while template switching is largely error-free. Template switching is frequently initiated at single-stranded (ss) DNA gaps in the wake of replication forks and proceeds via recombination-mediated sister chromatid junctions (SCJs) composed of pseudo-double Holliday junctions[6–9]. The importance of DDT is evident from highly increased skin cancer incidence in patients with mutations in components of the DDT machinery and recent observations that even a single unrepaired UV lesion can reduce the survival of budding yeast cells if they are deficient in DDT[10].

Although critical for the ability of cells to deal with genotoxic stress, DDT is a major source of mutations[4]. Significant progress has been made in uncovering the machinery that mediates DDT. This understanding has been focused on the identity of factors implicated in the two main DDT modes and the regulatory roles of PCNA mono and polyubiquitylation, which favor mutagenic TLS or largely error-free damage-bypass by template switching, respectively[11]. Besides template switching, a HR-dependent but PCNA polyubiquitylation-independent recombination pathway manifested through SCJs is deployed later in cell cycle or when PCNA SUMOylation is defective[8,12]. The latter DDT pathway is often referred to as the "salvage pathway" and can be activated genetically by inducing loss of PCNA SUMOylation, which physiologically declines upon completion of replication[8,13,14]. Genetic evidence further indicates that the activation of the salvage pathway by loss of PCNA SUMOylation can be recapitulated by loss of the Srs2 anti-recombinase, recruited by SUMOylated PCNA[13,14]. However, how the two recombination-dependent DDT pathways are regulated and the implications for genome integrity for using template switching versus salvage pathways are only partially understood[15–18].

DDK stands for the Dbf4-Cdc7 kinase and is required for origin firing and replication initiation[19]. Post-initiation of chromosome replication, DDK facilitates TLS-mediated mutagenesis in both budding yeast and mammalian cells[20–23]. Moreover, newly identified targets of DDK and DDK inhibition-induced fork instability suggested DDK roles in replication fork protection and restart[24–26]. Besides mutual regulations reported between DDK and checkpoint kinases[22,27,28], DDK engaged in replication is monoSUMOylated at multiple sites on both Dbf4 and Cdc7 subunits[29]. Similar SUMOylation events have been reported in human cells[30,31]. SUMO chain extension, prevented by the SUMO protease Ulp2, can trigger SUMO-targeted ubiquitin ligase Slx5/Slx8-mediated proteasomal-degradation of DDK engaged in replication[29] but the physiological function of monoSUMOylated DDK remains unknown.

Here, we investigated functions of DDK in replication-associated processes triggered by genotoxic stress. We uncover that DDK kinase activity and its monoSUMOylation promote recombination-mediated DDT and prevent replication fork uncoupling. The defects of DDK mutants in DDT and replication fork architecture associate with reduced levels of Rad51 proximal to damaged sites and are recapitulated by Rad51 mutants with point mutations in putative SUMO Interacting Motifs (SIMs) known to mediate interaction between Rad51 and SUMO in a manner dependent on Rad52[32,33]. The fork uncoupling defect of DDK mutants, but not the recombination-mediated gap-filling defect, is rescued by genetic activation of salvage recombination, thus uncovering the location and purpose of the salvage pathway in mitigating ssDNA accumulation at stalled replication forks. Altogether, the results reveal that monoSUMOylated DDK kinase facilitates Rad51-mediated gap-filling and prevents fork uncoupling during replication of damaged templates.

## Results

**DDK kinase facilitates template switch replication.** DDK is known to facilitate mutagenesis-mediated DNA damage bypass in budding yeast and mammalian cells[20,21,34]. We aimed to investigate if DDK also contributes to the recombination-mediated DDT branch. To this end, we used a *CDC7* allele, *cdc7-4*, in which the activity is abolished at the restrictive temperature of 37 °C causing lethality. Because DDK is essential for replication initiation[19,35], we first investigated experimental conditions that allow *cdc7-4* cells to perform the essential function and progress in the cell cycle as assessed by flow cytometry (Supplementary Fig. 1a). We conducted these experiments both in unperturbed conditions or in the presence of sub lethal doses of MMS that induce DDT and template switching[8]. We observed that keeping *cdc7-4* cells for 20 min at the permissive temperature of 25 °C after release from G1 arrest in the presence or absence of MMS allows progress into cell cycle (Supplementary Fig. 1b). As 25 °C is a permissive temperature, we experimented increasing the temperature of *cdc7-4* cells so that proliferation is not affected but at which repair defects may be manifested, without shifts of temperatures during the experiment. We found that 28 °C allows cell proliferation for *cdc7-4* (Supplementary Fig. 1c) but causes DNA repair defects (see below). To confirm that replication initiation/synthesis is not affected in *cdc7-4* cells grown in permissive conditions, we measured by ChIP-qPCR the ability of WT and *cdc7-4* cells to incorporate BrdU when released from G1 arrest in the presence of MMS at either 28 °C or 37 °C. While BrdU incorporation was almost absent in *cdc7-4* cells released from G1 arrest at 37 °C, as expected due to the arrest of cells in G1 (Supplementary Fig. 1c), similar BrdU uptake proximal to the ARS305 origin of replication was observed in *cdc7-4* and WT at 28 °C proximal to origins of replication (Supplementary Fig. 1c), thus indicating that, in this experimental condition, the replication initiation function of DDK is not affected by the *cdc7-4* mutation.

We next examined the effect of *cdc7-4* on replication-associated recombination events known as template switching. We combined *cdc7-4* with conditional inactivation of the Sgs1 helicase, the main activity required for the removal of SCJs mediating DNA damage bypass by recombination[36]. Sgs1 inactivation leads to SCJ accumulation composed primarily of pseudo-double Holliday junctions[9], enabling their visualization in 2D gel-DNA electrophoresis gels as X-shaped structures (Fig. 1a). To induce Sgs1 dysfunction without causing potential negative genetic interactions by deleting *SGS1* in *cdc7-4* cells, we employed the *Tc-sgs1* allele in which tetracycline addition causes destabilization of the mRNA and reduces HA-Sgs1 protein levels[37,38]. We observed a reduction in the levels of SCJ intermediates when *cdc7-4* mutation was expressed in cells conditionally depleted of Sgs1 (Fig. 1b).

To ascertain the role of the DDK kinase activity in facilitating replication-associated recombination, we further investigated, using *sgs1Δ* cells, the effect of the *cdc7-as3* allele, in which the kinase activity of Cdc7 can be conditionally inhibited by addition

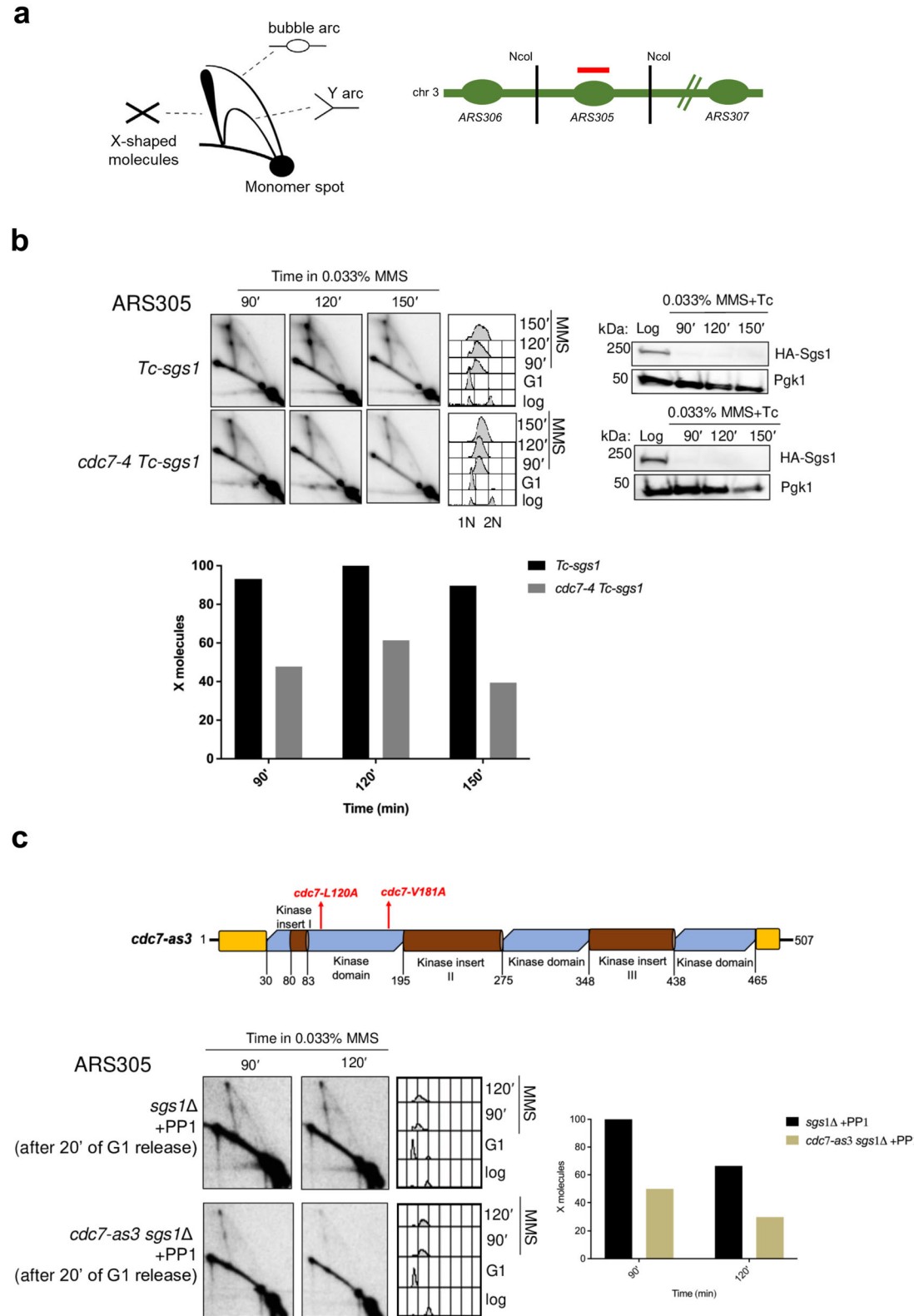

of the specific inhibitor PP1[39]. In this case, we added PP1 20 min upon release of cells from G1 arrest when the majority of cells were budded, to allow replication initiation and progression through S phase. We found a marked decrease in X-molecule formation upon pharmacological inhibition of DDK (Fig. 1c). Thus, using two conditions that partly affect Cdc7 activity without interfering with its essential function in DNA replication

initiation, we uncovered a role for DDK kinase activity in facilitating replication-associated recombination-mediated DDT.

**SUMOylated DDK promotes replication-associated recombination.** Previous work identified that DDK engaged in replication is monoSUMOylated at different residues on both Dbf4 and

**Fig. 1 DDK kinase facilitates template switch replication of damaged templates. a** Schematic representation of replication intermediates as visualized by 2D gel electrophoresis and of the ARS305 fragment spanned by NcoI restriction sites. **b** Exponentially grown *Tc-sgs1* and *cdc7-4 Tc-sgs1* cells were arrested in G1 phase at 25 °C using α-factor and released into S phase at 28 °C in the presence of 0.033% MMS and tetracycline (1 mM) to induce DNA damage and Sgs1 depletion respectively. Samples were collected at the indicated timepoints for 2D gel electrophoresis, FACS and western blotting analysis in two independent experiments. Collected cells were subjected to in vivo psoralen-mediated DNA inter-strand cross-linking followed by genomic DNA extraction and genome digestion with NcoI. 2D gel filters were hybridized with a probe recognizing the early replication origin, *ARS305*. HA-Sgs1 levels were detected by western blotting with Pgk1 as loading control. FACS profiles showing the cellular DNA content to monitor cell cycle progression are reported for each strain at the indicated time point. 1 N and 2 N cellular DNA contents below the FACS indicate G1 and G2/M phases of the cell cycle, respectively. The relative enrichment of X-shaped replication intermediates is represented in the quantified plots where signal intensities are normalized to the monomer spot and the highest value for the X molecule obtained during quantification was assigned as 100%. **c** Schematic representation of the *cdc7-as3* allele with respective mutations. *sgs1Δ* and *cdc7-as3 sgs1Δ* cells were synchronized in G1 phase with α-factor and released into YPD media containing 0.033% MMS. PP1 drug (20 mM) was added after 20 min from the release of cells from G1 into S phase. Samples for 2D gel analysis of replication-recombination intermediates and FACS analysis to detect the cellular DNA content and cell cycle phase were collected at the indicated time points and processed as described in panel **b**.

Cdc7 subunits[29]. The multi-monoSUMOylated DDK is protected by the Ulp2 SUMO protease against SUMO chains-buildup, which can cause proteasomal degradation of DDK. However, the functional significance of DDK monoSUMOylation remains unknown, as reduction of DDK SUMOylation does not affect the origin firing efficiency, the temporal program of replication or cell cycle progression[29]. Because several factors implicated in DNA damage bypass contain in their structure SUMO interacting motifs (SIMs) that can interact with SUMOylated residues in PCNA and potentially other DDT regulators[40], we tested the effect of SUMOylation defective DDK mutant *ddk-KR*, in which main SUMO sites have been mutated reducing the overall SUMOylation of DDK[29]. We found that *ddk-KR* reduced the level of replication-associated X-molecules (Fig. 2a). Moreover, mutations in the main SUMOylation site on Cdc7, K16R, conserved through evolution[29], also caused a decrease in X-shaped intermediates accumulating in Sgs1-depleted cells (Supplementary Fig. 2). Thus, monoSUMOylated DDK facilitates recombination-mediated DDT, potentially by attracting SIM-containing factors at DNA replication forks facing DNA lesions.

Many organisms and mammalian cells are diploid and could form besides SCJs, inter-homolog junctions (IHJs) during HR-mediated DDT and DNA repair. To analyze whether DDK defect in sister chromatid recombination is compensated by an increase in inter-homologous recombination events, we utilized a diploid strain in which one of the *ARS305* loci was edited to remove the adjacent *Eco*RV site on its left[41]. Due to this genomic editing, upon *Eco*RV and *Nco*I digestion, two *ARS305* restriction fragments of different lengths are formed (Fig. 2b). This heterozygosity allows distinct visualization of SCJs formed on each of the chromosomes III and the observation of less abundant IHJs (Fig. 2b)[41]. We genetically modified this system to homozygously express the *Tc-sgs1* conditional allele and either *cdc7-4* or *ddk-KR* alleles. We recapitulated the observation that SCJs are strongly reduced by *ddk-KR* and *cdc7-4* mutations in the absence of Sgs1, along with IHJs that were almost undetectable in the employed DDK mutations (Fig. 2b). These results thus uncover a role for DDK and its monoSUMOylation in supporting replication-associated recombination, without affecting the program of partner choice that involves a strong bias to the sister chromatid typical of mitotic recombination repair[41–43].

**SUMOylated DDK kinase prevents replication fork uncoupling.** Because DDK is associated with the replisome during replication elongation[24,27], we asked whether DDK mutations that impair recombination-mediated DDT affect replication fork architecture during DNA lesion bypass. Using in vivo psoralen-mediated DNA inter-strand crosslinking combined with low angle rotary shadowing and transmission electron microscopy[44,45], we

analyzed the fine ultra-structure of DNA replication intermediates. We performed this experiment starting from cells synchronously released from G1 arrest in MMS. We observed significant increase in gapped replication forks, defined as replication forks with a single-stranded (ss) DNA gap at the fork branching point larger than 200 nts, in both *cdc7-4* and *ddk-KR* mutants across three different experiments (Fig. 3a, b). Specifically, while gapped forks induced by MMS in wild type (WT) cells represented around 11% of the total replication intermediates scored, both *cdc7-4* and *ddk-KR* mutant cells accumulated around 33% of gapped forks in the same experimental conditions. To analyze if there is also a difference in the size of the single-stranded DNA (ssDNA) gap forming proximal to the replication fork branching point in *ddk* mutants versus wild-type (WT), we then focused on acquiring and analyzing gapped forks from three independent experiments to reach a larger number of such intermediates also in WT cells to allow statistical analysis. The result indicated that while the length of the ssDNA exposed proximal to the replication fork junction was distributed around 486 nts in WT cells, there was a marked increase in the length of the ssDNA gap in both *cdc7-4* and *ddk-KR* mutants where the length of the ssDNA discontinuities at the fork branching points were distributed around 721 and 735 nts, respectively (Fig. 3c). Of interest, these gapped forks do not impair bulk DNA replication, a result in agreement with recent ones in mammalian cells that report physiological tolerance to ssDNA arising during replication, as long as ssDNA is protected by the surplus of RPA molecules present in cells[46,47].

Next, we asked whether the increase in gapped forks was caused by deregulated nucleolytic activities[48]. Both Mre11 and Exo1 nucleases have been implicated in fork resection events in mammalian cells and budding yeast. Using *mre11-H125N* nuclease deficient and *exo1Δ* alleles, we found that ablation of either of these nuclease activities did not suppress the MMS-induced accumulation of gapped forks in *cdc7-4* cells (Fig. 3d). Moreover, the length of the MMS-induced ssDNA gaps at the fork branching points of *cdc7-4* cells was not altered by the inactivation of Mre11 and Exo1 nuclease activities (Fig. 3e). Altogether, these results indicate that the increased gapped forks in DDK mutants represent uncoupling events generated during DNA replication of damaged templates rather than resection of the nascent strands of replication forks by nucleases.

**MonoSUMOylated DDK promotes Rad18/Mms2-dependent gap-filling.** To investigate the mechanism by which DDK promotes recombination-mediated DDT, we first combined *cdc7-4* with deletions in *RAD18* and *MMS2*, known to mediate PCNA mono- and polyubiquitylation and to be critical for SCJ formation during template switching[8,49]. The effect of the *cdc7-4* mutation

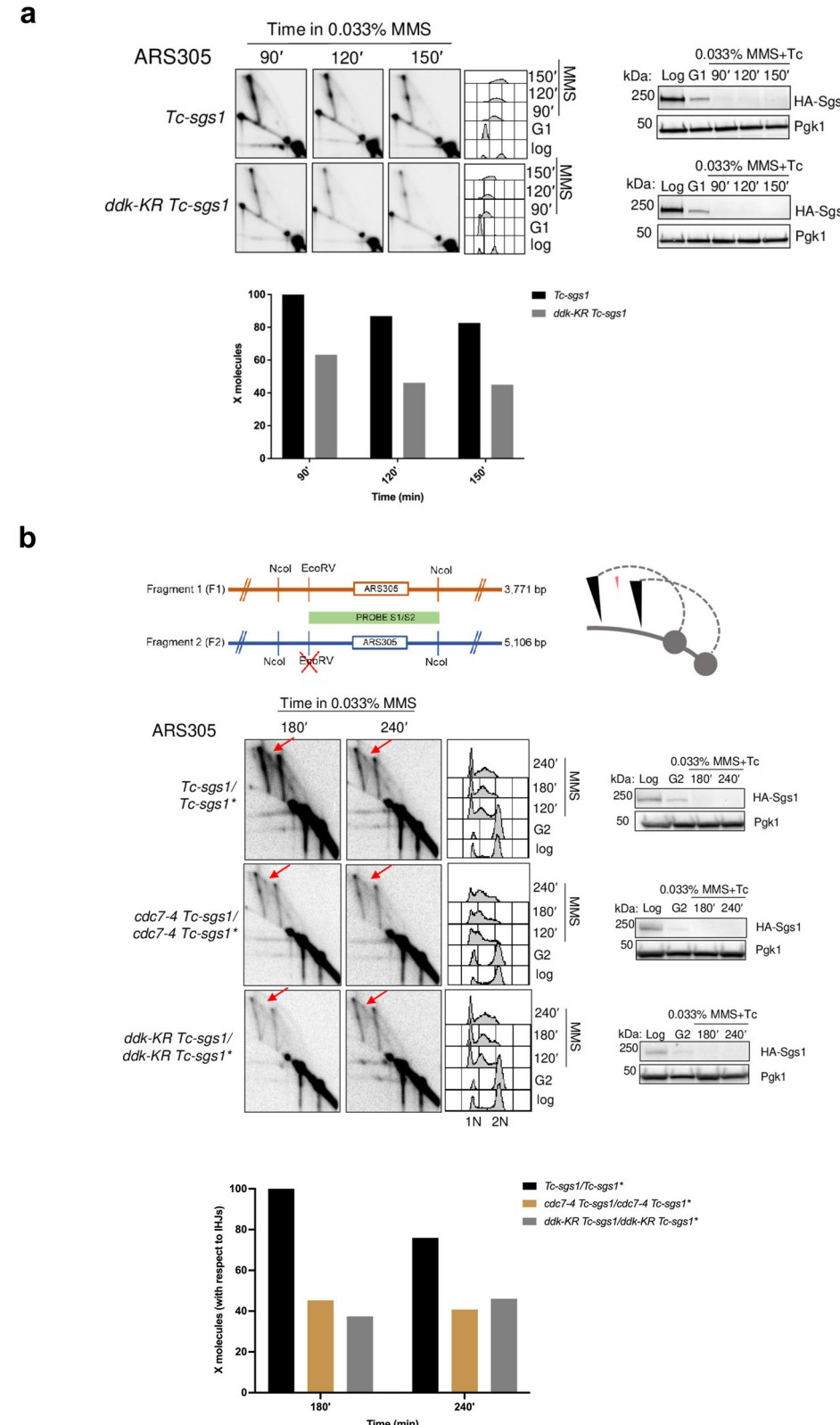

in reducing the damage-dependent SCJ accumulation in the absence of Sgs1 was similar, although less pronounced, to *rad18Δ* and *mms2Δ* mutations. Moreover, the combination of *cdc7-4* with *RAD18* and *MMS2* deletion did not further aggravate the defect in SCJ formation observed in the absence of *SGS1* (Fig. 4a and Supplementary Fig. 3a), suggesting joint roles between DDK and PCNA polyubiquitylation in template switching.

We observed additive defects between *rad18Δ*, *mms2Δ* and *cdc7-4* mutations for MMS sensitivity (Supplementary Fig. 3b). As *cdc7-4* is defective in TLS-mediated damage bypass[20] and both mutagenic and recombination modes contribute to damage tolerance as assessed in cell viability[50], we attribute the aggravation of the MMS sensitivity in the double mutants to the dysfunction in mutagenic bypass of *cdc7-4* that is not shared

**Fig. 2 MonoSUMOylated DDK promotes replication-associated recombination. a** *Tc-sgs1* and *ddk-KR Tc-sgs1* cells were synchronized in G1 phase at 25 °C and released in YPD medium containing 0.033% MMS and tetracycline (1 mM) at 28 °C. Samples were then analyzed by 2D gel electrophoresis in two independent experiments. Experimental setup is the same as described in Fig. 1a. **b** (Top) Schematic representation of the 2D gel patterns arising from the modified diploid strain with the expected migration of inter-homolog Junctions (IHJs) in red versus sister chromatid junctions in black. IHJs are indicated by red arrows. (Bottom) *Tc-sgs1/Tc-sgs1*\*, *cdc7-4 Tc-sgs1/cdc7-4 Tc-sgs1*\*, *ddk-KR Tc-sgs1/ddk-KR Tc-sgs1*\* (\* indicates mutation at the *EcoRV* restriction site adjacent to the ARS305 region) were arrested in G2/M phase with nocodazole at 25 °C and released at 28 °C in YPD medium containing 0.033% MMS and tetracycline (1 mM) to induce Sgs1 depletion. Samples were collected at the indicated timepoints which were psoralen crosslinked and genomic DNA was extracted for 2D gel analysis in two independent experiments. Restriction digestion was performed with *EcoRV* and *NcoI* and the resulting fragments separated by 2D gel electrophoresis and later probed for ARS305. HA-Sgs1 depletion was visualized using an HA antibody with Pgk1 as loading control. Cell cycle progression of each strain was monitored by FACS analysis. The relative enrichment of X-shaped replication intermediates is represented in the quantified plots where signal intensities from IHJs were normalized with respect to the monomer spot and the highest value for the X molecule obtained during quantification was assigned as 100%.

---

by Mms2 and Rad18. Of interest, *ddk-KR* did not induce sensitivity to MMS and, while it did not aggravate the sensitivity of *rad51Δ* cells (Fig. 4b), it suppressed the sensitivity associated with loss of Mms2 (Fig. 4c) and very mildly that of *rad18Δ* (Supplementary Fig. 3c). Together with the effects of SUMOylated DDK in mediating replication-associated recombination (Fig. 2), these results indicate DDK monoSUMOylation as a regulator of the gap-filling process through template switching mediated by the joint action of Rad51-dependent recombination and PCNA polyubiquitylation.

**MonoSUMOylated DDK kinase promotes salvage recombination.** We next asked whether DDK is important for the salvage pathway of recombination. Like template switching, the salvage pathway is activated by DNA damage but proceeds independently of PCNA mono- and polyubiquitylation[16]. The salvage pathway can be genetically activated by the *siz1Δ* mutation, which reduces PCNA SUMOylation, and subsequently decreases the recruitment of the anti-recombinase Srs2 to sites of lesions, allowing Rad51 nucleofilament formation even in potentially toxic contexts[18]. To address if similar to Rad51 and the HR machinery[8,12], DDK plays a role also in the salvage pathway, we addressed the contributions of *cdc7-4* and *ddk-KR* mutations towards SCJ formation in a *Tc-sgs1 siz1Δ* background. We found that both *cdc7-4* and *ddk-KR* mutants significantly reduced the damage-dependent SCJs arising in the context of salvage recombination activation (Fig. 5a).

Since all the recombination-mediated DDT mechanisms rely on Rad51 recruitment to damaged sites, we decided to use a ChIP-qPCR-based approach to test whether mutations in DDK affect Rad51 recruitment/retention proximal to active DNA replication origins, reported as sites of DNA replication fork stalling and damage bypass[51]. We found that the levels of Rad51 associated with stalled forks is reduced in both *ddk-KR* and *cdc7-4* mutants (Fig. 5b). Thus, DDK kinase activity and its mono-SUMOylation facilitate the recruitment and retention of Rad51 proximal to damage sites to favor recombination-mediated DDT.

**Salvage pathway activation rescues fork uncoupling.** The nature of the salvage pathway and the context in which it occurs is not currently understood[16]. Because DDK mutants are characterized also by an increase in gapped replication forks upon DNA damage (Fig. 3a, b), we asked if activation of the salvage pathway by loss of Siz1 influences this phenotype. To this end, we conducted TEM experiments to analyze DNA replication fork architecture also in the *siz1Δ* background. While *siz1Δ* mutants behaved similarly with WT in terms of replication fork architecture, the increase in gapped forks characteristic of DDK mutants was completely suppressed by *siz1Δ* (Fig. 5c). Deletion of *SIZ1* also reduced the length of the ssDNA gaps accumulating at the branching point of the forks of *ddk-KR* cells but not the one of

*cdc7-4* cells (Fig. 5d). The effect of *siz1Δ* on counteracting accumulation of gapped forks and long ssDNA stretches at the fork branching points was recapitulated by loss of Srs2 (Supplementary Fig. 4a, b). Taken together, these results indicate that activation of the salvage pathway prevents fork uncoupling during replication of damaged templates and reveal that the salvage pathway takes place preferentially at stalled replication forks rather than at gaps left behind replication forks.

**Rad51 putative SIMs mediate SCJs and prevent fork uncoupling.** Because SUMOylated DDK is critical for SCJ formation and facilitates Rad51 enrichment at damaged replication forks, we reasoned that the underlying mechanism involves engagement between SUMO interacting motifs (SIMs) of HR factors with SUMOylated residues on DDK. Previous work identified a putative SIM in RAD51, required for accumulation of mammalian RAD51 at sites of DNA damage[52]. Two putative SIM motifs (277IVV279 and 321VVV323) are present in budding yeast Rad51 (Supplementary Fig. 5a), with their mutations shown to abolish the interaction with SUMO[32]. The second SIM was further characterized[33], revealing that it is required for robust interaction with Rad52, but not affecting the one with Rad54. However, the potential role of these motifs in Rad51 in replication-associated recombination remains to date unknown.

We engineered mutations in putative SIM1 and SIM2 by replacing the three critical residues with alanine (Supplementary Fig. 5a), as previously reported[32]. The mutations caused MMS hyper-sensitivity similarly with *rad51Δ*, but only partial sensitivity to Zeocin that causes DSBs (Fig. 6a). While mutations in the putative SIMs of Rad51 did not interfere with protein stability (Supplementary Fig. 5b), they drastically diminished the accumulation of Rad51 on damaged chromatin proximal to replication regions (Supplementary Fig. 5c). Moreover, the mutations in the putative SIMs of Rad51 nearly abolished DNA damage-induced recombination-mediated SCJ formation as detected by 2D gels in the *sgs1Δ* mutant background (Fig. 6b). Next, we addressed whether the putative Rad51 SIMs also influence the replication fork architecture. To date, no analysis of how replication fork structures are affected by Rad51 loss upon chronic exposure to MMS has been reported in budding yeast. EM analysis revealed that both Rad51 loss and mutations in the putative SIMs of Rad51 cause an increase in gapped replication forks (Fig. 6c), resembling the increase in gapped forks in *ddk* mutants. Both *rad51-SIM* mutants and *rad51Δ* also displayed an increase in the length of ssDNA exposed at replication fork junction points upon DNA damage (Fig. 6d). Thus, we uncovered that Rad51 and its putative SIMs play a major role in mediating recombination-dependent damage bypass and in preventing fork uncoupling upon genotoxic stress. However, the exact role of the motifs identified as putative SIMs in modulating Rad51 functionality remain to be explored by future studies.

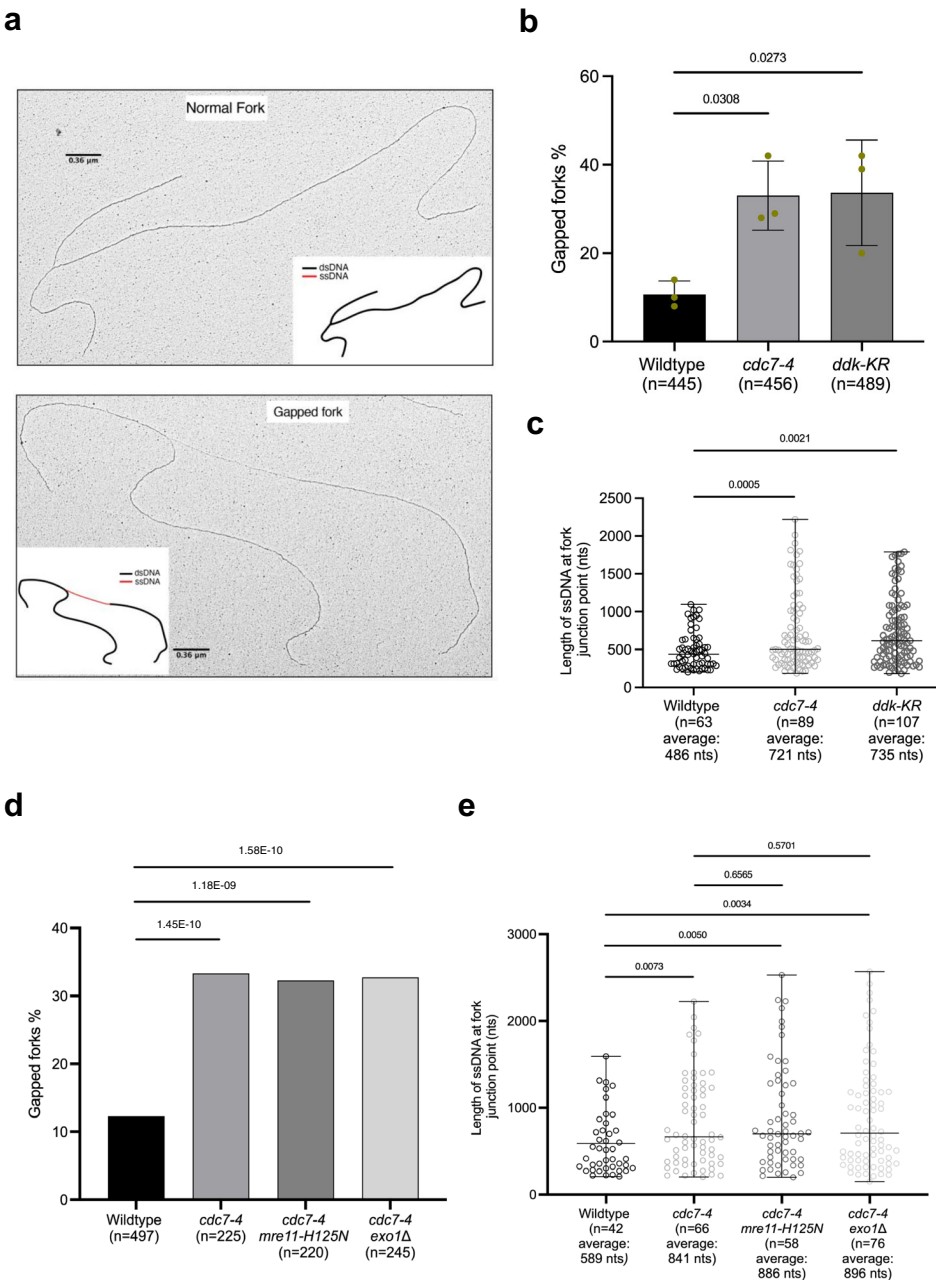

**Fig. 3 SUMOylated DDK kinase prevents replication fork uncoupling. a** Examples of normal and gapped forks visualized by Transmission electron microscopy (TEM) analysis along with schematic representations of DNA molecules with dsDNA (black) and ssDNA (red). Scale bars (black) of 360 nm calculated for dsDNA are reported. **b** TEM analysis of replication intermediates. Exponential cultures were arrested in G1 phase and released into S-phase in medium containing 0.033% MMS at 28 °C for 60 min. Histogram reports the percentages of gapped replication forks, from three independent experiments. The gapped forks category contains a small fraction of broken forks, derived from the mechanical breakage of the long ssDNA stretches present at the branching point of the gapped forks. The total number of replication intermediates scored in the three experiments is indicated as '*n*'. Error bars represent standard deviation of three independent experiments, the center of the bar represents the average value. *P*-values were calculated by ordinary one-way ANOVA test. **c** Scatter dot plot representing the distribution of lengths of ssDNA stretches at the branching points of forks of indicated cells. Values are derived from the analysis of the samples presented in **b**. Central line in the plot indicates the average length of ssDNA discontinuity (in nucleotides), the bars represent distribution range. *P* values were calculated by unpaired two-tailed Student's *t* test. **d** TEM analysis of the DNA replication intermediates formed in the same experimental conditions described in **b** in two independent experiments. '*n*' represents the total number of DNA replication intermediates analyzed in the indicated sample. Histogram represents the percentage of gapped forks in the indicated cells as described in **b**. *P* values were calculated by Fischer's exact two-sided test. **e** Scatter dot plot representing the distribution of the lengths of the ssDNA stretches (in nucleotides) at the fork branching points of the indicated yeast strains. '*n*' is the total number of DNA replication intermediates analyzed. The middle line in the scatter dot plot indicates the average length of the ssDNA discontinuity (in nucleotides), bars show the distribution range. *P*-values were calculated by unpaired two-tailed Student's *t* test.

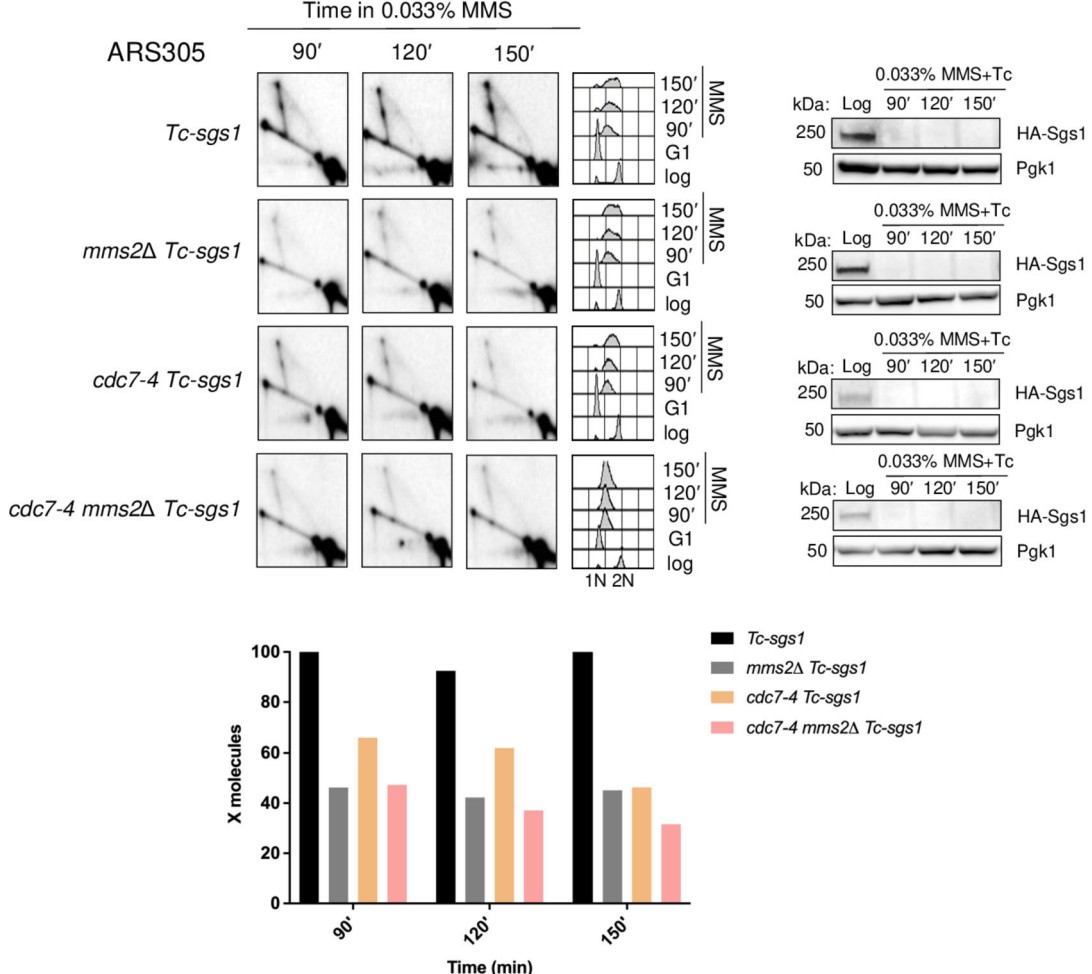

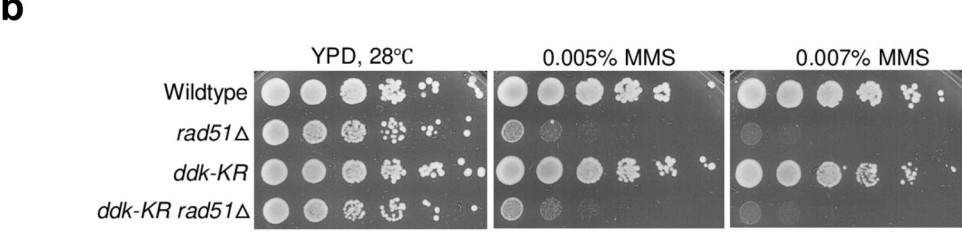

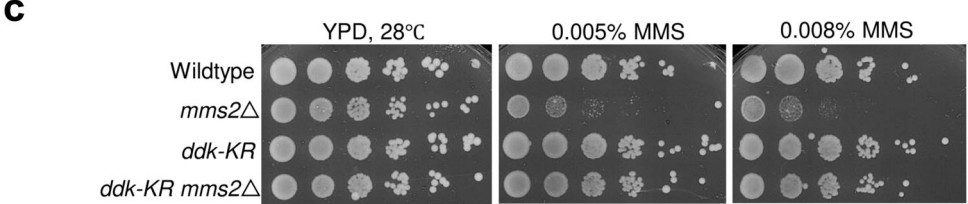

## Discussion

DDK kinase, built up of the Cdc7 kinase and the Dbf4 regulatory subunit, is well known for its essential role in replication initiation by phosphorylating and activating the replicative MCM helicase[19,53]. Recent studies have indicated that DDK phosphorylates several other substrates during the elongation stage of DNA replication and upon replication stress, although DDK functions in these latter contexts are less understood[24,27]. In response to mutagens, DDK facilitates DNA lesion bypass via TLS polymerases. In vertebrate cells, upon UV irradiation, DDK phosphorylates the RAD18 ubiquitin ligase, facilitating its chromatin binding and subsequent PCNA ubiquitylation, which attracts DNA polymerase η for TLS[21,23]. A role of DDK in activating TLS was observed initially in budding and fission yeast, in

**Fig. 4 SUMOylated DDK promotes Rad18- and Mms2-dependent gap-filling. a** *Tc-sgs1, mms2Δ Tc-sgs1, cdc7-4 Tc-sgs1, cdc7-4 mms2Δ Tc-sgs1* cells were synchronously released from G1 phase into S-phase in medium containing 0.033% MMS and 1 mM tetracycline at 28 °C in two independent experiments. Samples were collected at the indicated timepoints and subjected to in vivo psoralen-mediated DNA inter-strand crosslinking followed by genomic DNA extraction, digestion with *Nco*I, 2D gel electrophoresis and probing of the 2D gels filters with the ARS305 probe. HA-Sgs1 depletion was monitored by western botting using Pgk1 as control. FACS plots show cell cycle progression. The signal of X molecules was quantified with respect to the monomer spot. The highest valued obtained for the signal of X molecules was considered as 100%. **b, c** Exponentially growing cultures of the indicated strains were adjusted to the same concentration, serially diluted 10-fold and spotted on YPD plates containing the indicated MMS concentrations. Plates were incubated at 28 °C and photographed after 2 days from the spotting in two independent experiments with similar results.

which DDK mutants are defective in UV and MMS-induced mutagenesis[54–57], whereas high levels of Cdc7 in budding yeast cause hypermutability in response to UV[58]. Moreover, studies in budding yeast identified a physical interaction between Cdc7 and the Rev7 subunit of Pol ζ, and, based on genetic analysis of mutations obtained in a reversion assay, Cdc7 was shown to contribute to Pol ζ-dependent and independent large deletions arising during postreplicative gap-filling[34].

Here, we sought to investigate roles for DDK in replication-associated recombination. The premise for such a function resided in the fact that DDK is important in meiosis for recombination and its coordination with replication[59–61]. However, DDK roles in mitotic recombination have not been documented. Several previous results assigned DDK to the *RAD6/RAD18* epistasis group[56], comprising both TLS and HR branches of DDT, and revealed partially non-overlapping functions of DDK with other TLS-mediated branches[20]. Moreover, defects in inducing mutagenesis upon DNA damage observed in DDK mutants have been reported in 9-1-1 mutants[62,63], subsequently shown to facilitate the HR-mediated mode of damage-bypass[12]. Notably, non-redundant roles of DDK with ubiquitinated/SUMOylated PCNA in gap-filling, inferred based on sequence patterns of complex mutations[34], could be accounted in principle by a role of DDK in the recombination-mediated mode of DDT independently of PCNA regulation by ubiquitin and SUMO.

Using several DDK mutations in experimental conditions that do not impair origin firing, here we uncovered a role for DDK kinase in promoting replication-associated recombination. Importantly, besides the role of the DDK kinase activity, we revealed a critical contribution of DDK monoSUMOylation in this process (Fig. 7). Reduction in DDK SUMOylation or activity did not affect the program of mitotic recombination partner choice that involves a strong bias to the sister chromatid. This program of donor choice is enforced by sister chromatid cohesion[43], which DDK supports via its role in facilitating cohesin loading at centromeres[64,65]. Altogether, these results indicate that the underlying mechanism by which DDK facilitates replication-associated recombination lies elsewhere. We observed roles of DDK kinase and its monoSUMOylation in facilitating recombination-mediated damage bypass in backgrounds useful to monitor both template switching, relying on PCNA poly-ubiquitylation and HR, and salvage recombination, which is independent of Rad18 and PCNA polyubiquitylation but relies on HR[16] (Fig. 7). Based on the pattern of the observed defects, we concluded that monoSUMOylated DDK kinase must affect the functionality of factors associated with the core HR machinery. One potential mechanism is by engaging SIMs present in their structure. Rad51 comprises putative SIMs conserved through evolution[32,33] shown to affect Rad51 interaction with SUMO and its localization to damaged sites in mammalian cells[52]. We discovered that point mutations in the two putative SIMs of yeast Rad51 preserved protein stability but resulted in severe phenotypes, close to complete loss of Rad51 in what regards MMS-associated phenotypes and Rad51 recruitment to replicating regions, but partial in what regards Zeocin sensitivity. The

replication-associated recombination defects resembled the ones of DDK SUMOylation defective mutants but were more severe, indicating SUMOylated DDK as a regulator of the core Rad51 function in the recombination-mediated replication restart and gap-filling (Fig. 7). Moreover, the exact functionality of the motifs encoding the putative SIMs will require further biochemical characterization. However, as we did not detect physical interaction between DDK and Rad51 using co-immunoprecipitation experiments, it is possible that such interaction is mediated by the MCM helicase, which interacts with Rad51[66] and is a target of DDK[19]. Alternatively, it is possible that the DDK regulation of the Rad51 function we uncovered here is mediated by other mechanisms.

Because DDK is associated with the elongating replisome, we investigated potential consequences of DDK dysfunction in terms of replication fork progression and architecture in the presence of MMS-induced DNA damage. We did not detect gross consequences on the ability of cells to proliferate and proceed through S phase in our experimental conditions. However, mild dysfunctions in the DDK kinase activity and partial reduction in DDK monoSUMOylation resulted in marked increase in replication forks containing large ssDNA stretches at their fork branching points (Fig. 7). These intermediates were not due to deregulated resection mediated by Exo1 or Mre11 nucleases, which are coordinated in part by DDK in mammalian cells[26,28], suggesting that the gapped replication fork intermediates analyzed in this study most likely arise in consequence of fork uncoupling events generated during replication of damaged templates.

The question we next addressed was whether such fork uncoupling events require the Rad51 recombinase. The functional link between DDK kinase, SUMOylated DDK and Rad51 in preventing fork uncoupling during replication across damaged sites was supported by our finding of reduced levels of Rad51 recruitment at stalled forks in *cdc7-4* and *ddk-KR* mutants. Using *RAD51* deletion and point mutants in the putative SIMs of Rad51, we uncovered that complete loss of Rad51 or dysfunctional interaction of Rad51 with SUMO phenocopies DDK mutants regarding replication fork architecture during replication of damaged sites. These results indicate that the SUMOylated DDK acts together with Rad51 in a functional axis critical to prevent fork uncoupling during replication in the presence of DNA damage.

Second, we asked if partial activation of a salvage pathway of recombination through reduced recruitment of the Srs2 anti-recombinase to SUMOylated PCNA at replicating regions[13,14] would alleviate the fork uncoupling phenotype caused by DDK mutations. This was indeed the case, revealing one biological scope of the salvage recombination pathway in limiting fork uncoupling and exposure of large ssDNA stretches proximal to the junction points of stalled forks (Fig. 7). Notably, because the same genetic activation of the salvage recombination pathway did not alleviate the defect in HR-mediated gap-filling associated with DDK mutants, altogether, the results presented here reveal that the salvage pathway of recombination generally acts at ssDNA

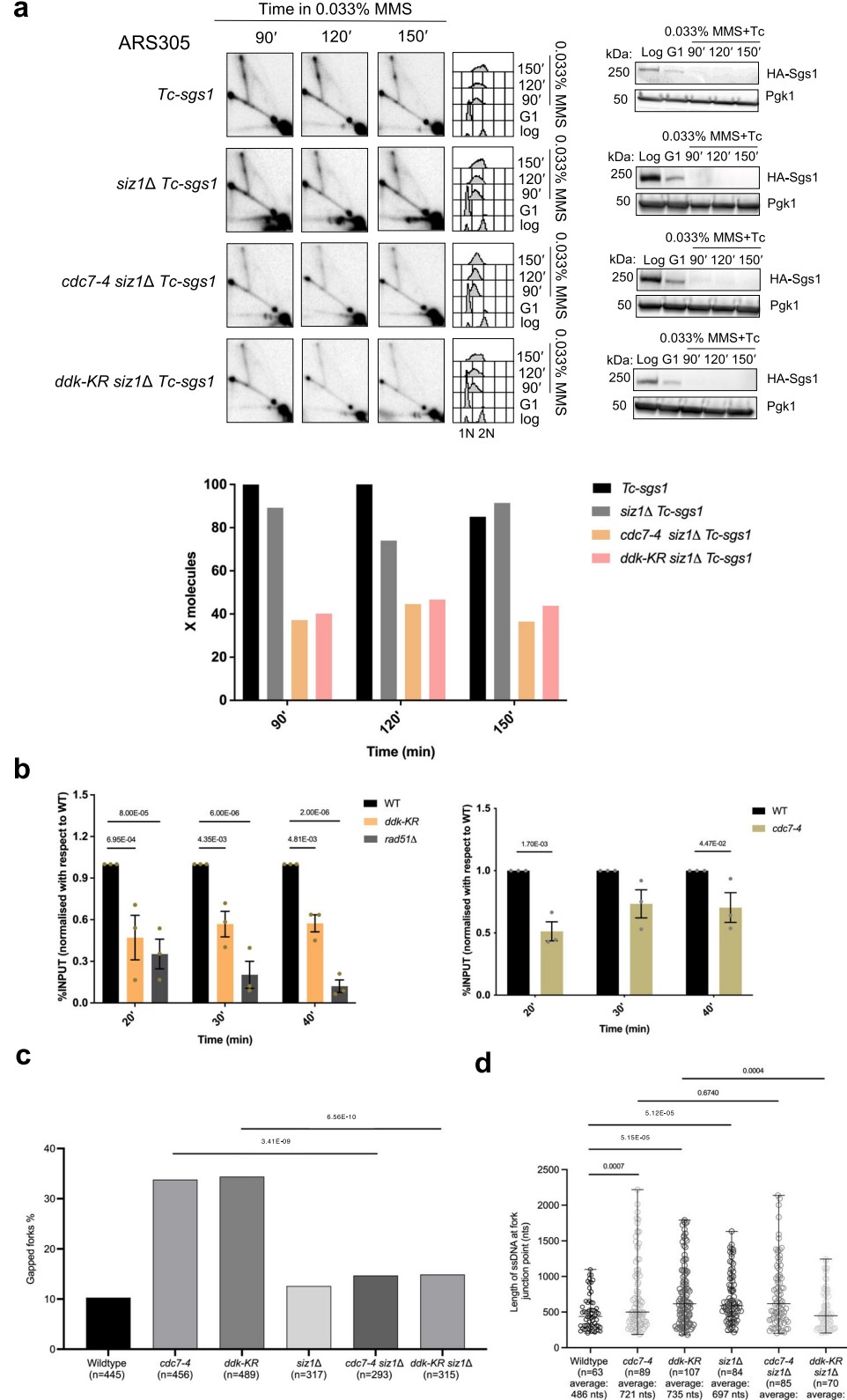

stretches proximal to the replication fork branching point, rather than at gaps confined behind replication forks (Fig. 7). The location is important, as activation of HR at stalled forks, which are prone to breakage and collapse due to exposure of large stretches of ssDNA, may cause genomic rearrangements[67]. The ensuing genomic instability may be due to error-prone DSB repair pathways activated by the broken forks[68] or erroneous invasions in the non-replicated region ahead of the fork or using donors with limited homology[69–72]. In contrast, gap-filling behind the replication fork typical of template switching[7,9] is likely more accurate due to the confined topology of the gap-region and potentially easier homology search[18,73].

Increased DDK expression in human cancers correlates with increased chemoresistance and higher mutation frequencies[74].

**Fig. 5 SUMOylated DDK facilitates salvage recombination that prevents fork uncoupling. a** 2D gel analysis of MMS-induced DNA replication-recombination intermediates accumulated in the indicated strains in two independent experiments. Exponentially growing cells were synchronized in G1 and released in YPD medium containing 0.033% MMS and tetracycline (1 mM) at 28 °C. Samples were collected at the indicated timepoints for FACS, western blotting, and 2D gel analysis. Experimental setup is the same as described in Fig. 1a. **b** Rad51 recruitment at the ARS305 replication origin was analyzed using ChIP-qPCR. Exponentially growing cells were arrested in G1 phase at 25 °C and released into 0.033% MMS-containing YPD medium at 28 °C with samples for ChIP-qPCR collected at the indicated time points. Each ChIP was repeated three times and each real time PCR was performed in triplicates. Histogram reports the enrichment signals expressed as percentage of the signal in the corresponding input. Enrichment values obtained were then normalized and expressed as fractions of the enrichment obtained in the wild type cells, which was considered as 1. Error bars are represented as SEM (mean value ± standard error of mean) of three independent experiments. The indicated *p*-values were calculated by two-way ANOVA test. **c** TEM analysis of the DNA replication intermediates in the indicated strains exposed to MMS. Experimental setup is the same as described in Fig. 3b. Histogram represents the percentage of gapped replication forks generated in *cdc7-4 siz1Δ* and *ddk-KR siz1Δ* mutants in comparison with *siz1Δ*. WT, *cdc7-4* and *ddk-KR* values are indicated for reference (see Fig. 3). Total number (*n*) of DNA replication-recombination intermediates analyzed for each genotype is indicated. The indicated p-values were calculated by Fischer exact test two-sided. **d** Scatter dot plot represents the distribution of length of the ssDNA gaps (in nucleotides) at the fork branching points of DDK mutants in comparison with WT. Analysis was conducted on the same samples of the experiment presented in **c**. Middle line in the scatter dot plot indicates the average value of the length of the ssDNA discontinuity, bars represent distribution range. *P*-values were calculated by unpaired two-tailed Student's *t* test are indicated.

Based on the current results and previously reported analysis of mutation patterns in budding yeast cells, we propose that the genomic alterations and increased mutations are compounded by upregulated TLS activities and usage of recombination pathways that can induce genomic alterations when employed during chromosome replication[69,72,75].

Altogether, we identified here a critical role for mono-SUMOylated DDK kinase in regulating Rad51 function during HR-mediated gap-filling and in preventing extensive fork uncoupling during DNA replication across lesions. The mechanism by which the latter is achieved requires further investigation. Regarding the effect of DDK on the Rad51 recombinase at stalled forks and during gap-filling, it is possible that DDK may enhance the functionality of motifs previously identified as putative Rad51 SIMs, which we find to be critical for Rad51 functions in conditions of replication stress, or counteract negative regulators of Rad51, such as Srs2. Depending on the availability of other recombination regulators, Rad51 filaments may favor Rad51 interaction with Polα-Primase to mediate repriming and subsequent DDT-mediated gap-filling events[6,7,76] or be engaged in recombination events at stalled replication forks, in a manner that can generate rearrangements[70] (Fig. 7).

Importantly, our findings delineate monoSUMOylated DDK kinase as a critical regulator of replication-associated recombination acting in complementary fashion with PCNA SUMOylation. Differently from PCNA SUMOylation, which is confined to early stages of replication and is relevant for regulating TLS and template switching[13,14,77,78], monoSUMOylated DDK promotes both template switch-mediated gap-filling and the salvage pathway of recombination, the latter generally activated after the bulk of replication is complete[8,12]. As monoSUMOylation of DDK is conserved across species[29,30] and RAD51 plays roles in fork protection in mammalian cells[48], it is likely that the mono-SUMOylated DDK and the RAD51 axis is critical also in mammalian cells for activating recombination at stalled replication forks. Such activation will counteract replication fork uncoupling and consequent replication fork breakage but may cause genomic rearrangements. Our findings can explain the complex pattern of mutations observed in cancers overexpressing DDK[74] and inspire future research on conserved mechanisms of DDT regulation at stalled replication forks.

## Methods

**Yeast strains**. The *Saccharomyces cerevisiae* strains used in this study are derivatives from W303 and the relevant genotypes are indicated in Supplementary Table 1. All the relevant mutants were constructed by genetic crosses, transformation, site-directed mutagenesis and standard techniques[79]. All genetic editions were verified by selection markers, PCR, and sequencing.

**Reagents and oligonucleotides**. The reagents and oligonucleotides used in this study are indicated in Supplementary Table 2, along with their source and identifier, where appropriate.

**Yeast culturing and synchronization**. Yeast strains were grown in YPD medium containing 2% glucose as carbon source at 25 °C. For cell cycle synchronization, exponentially grown cells at 25 °C were synchronized in G1 phase by adding alpha factor (Genscript) to a final concentration of 3–5 mg/ml for 2 h or in G2/M phase with Nocodazole to a final concentration of 10 μg/ml together with DMSO to a final concentration of 1% for 2 h at 25 C. The cells were released from the respective arrests by washing them with YP medium (YP + 1%DMSO in case of Nocodazole arrest) followed by its release into fresh YPD medium containing 0.033% MMS at 28 °C. To inhibit de-novo Sgs1 translation in cells expressing pADH1-Tc-3HA, tetracycline (NZYTech) was added to a final concentration of 1 mM at the indicated times. For drug sensitivity assays, cells from overnight cultures were counted and diluted before being spotted on YPD plates containing the indicated concentrations of drugs and incubated at 28 °C for 2–3 days.

**Flow cytometry**. FACS analysis was performed according to the protocol described in Fumasoni et al.[6] using SYTOX green staining (Invitrogen). For flow cytometric analysis, ~1–2 × 10^7 cells for each timepoint were collected and permeabilized in 70% ethanol. The cells were then treated with 200 μl of 50 mM Tris-HCl containing 2 mg/ml of RNase A (Sigma-Aldrich) for at least 3 h to overnight at 37 °C. After RNase treatment, cells were pelleted again and resuspended in 200 μl of 50 mM Tris-HCl containing 1 mg/ml of Proteinase K (Roche) and incubated at 50 °C for 30 min. Subsequently, cells were stained in SYTOX green solution (1 μM) (Invitrogen) and analyzed using a FACSCalibur™ Flow Cytometer for FL1H fluorescence and the BD CellQuest Software.

**Trichloroaceticacid protein extraction**. In all, 15 ml of 1 × 10^7 yeast cells were harvested by centrifugation and were resuspended in 100 μl of 20% trichloroaceticacid (TCA). Equal volume of glass beads (Sigma-Aldrich) was added and the suspension was vortexed for about 20 min for cell lysis. Later, 200 μl of 5% TCA was added to this mixture, to get a final concentration of 10% TCA, following the transfer of cell lysate to a fresh tube. The pellet of the proteins was obtained by centrifugation at 5000 rpm for 10 min and resuspended in 100 μl of 2X Laemmli buffer (4% SDS, 20% glycerol, 10% 2-mercaptoethanol, 0.004% bromophenol blue, 0.125 M Tris-HCl; pH 6.8). The suspension is neutralized by the addition of 50 μl of 1 M Tris base. The samples were boiled at 95 °C for 7 min and then centrifuged at 15,000 rpm for 5 min. After centrifugation, the supernatant containing the proteins were analyzed by western blotting. BIORAD Image Lab Version 5.2.1 for Western Blot Acquisition was used. The antibodies and their dilutions were as follows: anti-Rad51 (Santa Cruz sc-33626, 1: 2000), anti-HA (Invitrogen 12CA5, 1: 2000), amti-Pgk1 (Invitrogen, 22C5D8, 1:5000).

**ChIP-qPCR**. Chromatin immunoprecipitation (ChIP) was carried out as previously described[80]. Briefly, cells were arrested in G1 with α-factor and released at 28 °C in media containing 0.033% MMS. They were collected and crosslinked with 1% formaldehyde for 30 min. To stop the crosslinking, 2 ml of 2.5 M glycine was added to this solution and rotated for 15 min. Cells were then washed twice with ice-cold 1X TBS, resuspended in lysis buffer supplemented with 1X EDTA-free complete cocktail, and lysed using FastPrep-24 (MP Biomedicals). Chromatin was sheared to a size of 300–500 bp by applying 5 sonication cycles for 15 s at 20% power. After each round of sonication, the chromatin is pelleted by centrifugation at 2300 g for 1 min at 4 °C. In total, 10 μl of this supernatant (now considered as input) is added into another 1.5 ml centrifuge tube containing 190 μl of TE buffer with 1% SDS. IP

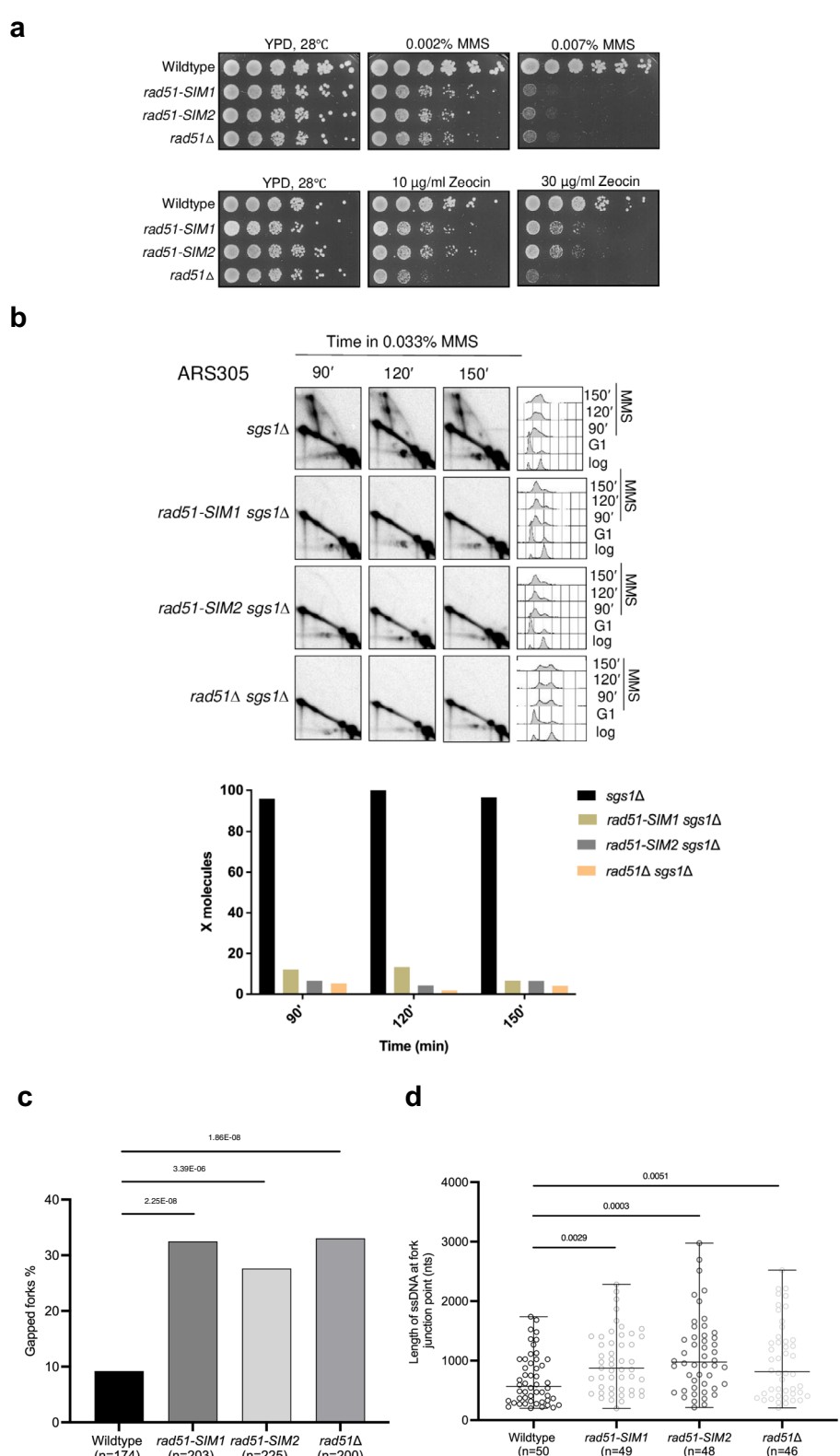

reactions with 0.8 mg anti-Rad51 antibody (Santa Cruz, sc 33626) and Dynabeads protein G were allowed to proceed overnight at 4 °C. After washing and eluting the ChIP fractions from beads, crosslinks were reversed at 65 °C overnight for both Input and IP. After proteinase K treatment, DNA was precipitated using sodium acetate (3 M) and 100% ethanol. Real-time PCR was performed using QuantiFast SYBR Green PCR kit according to the manufacturer's instructions and each ChIP experiment was repeated 3 times and each qPCR was performed in triplicates using a Roche Light Cycler 96 system and Numbers software as previously reported[37].

The results were analyzed with absolute quantification/2nd derivative maximum and the 2(-delta(t)) method. Statistical analysis was performed using 2-way ANNOVA. The error bars represent standard error of mean (SEM).

**BrdU qPCR**. BrdU qPCR was carried out largely as described in Psakhye et al.[29]. In total, 150 mL of exponentially cell culture was synchronized in G1, released into S-phase in the presence of 0.033% MMS and 200 μg/ml BrdU. Cells were harvested at

**Fig. 6 Rad51 and putative SIMs mediate recombination-mediated damage bypass and prevent fork uncoupling. a** Exponentially growing cultures of the indicated strains were adjusted to the same concentration, serially diluted (1:10), and then spotted on YPD plates with the indicated drug concentrations in two independent experiments. Plates were photographed after 3 days of spotting. **b** *sgs1Δ*, *rad51-SIM1 sgs1Δ*, *rad51-SIM2 sgs1Δ*, and *rad51Δ sgs1Δ* cells were synchronized in G1 phase and then released into 0.033% MMS containing YPD medium. Samples were collected for the indicated timepoints for FACS, western blot, and 2D gel analysis in two independent experiments. Collected samples for 2D gel analysis were subjected to in vivo psoralen crosslinking, after which genomic DNA was extracted followed by *NcoI* digestion and analysis of 2D gel electrophoresis-resolved intermediates for the ARS305 region. The relative enrichment of X-shaped replication intermediates was quantified as described in Fig. 1c. **c** Electron microscopic analysis of wildtype, *rad51-SIM1*, *rad51-SIM2*, and *rad51Δ*. Cells were arrested in G1 phase using α-factor and then released into medium containing 0.033% MMS. Samples were collected at 60 min and subjected to in vivo psoralen crosslinking. Genomic DNA was extracted and subjected for electron microscopy analysis. The histograms indicate the percentage of gapped replication forks observed for the above strains. The total number of forks analyzed for each strain is reported as 'n'. The P-values were calculated by Fischer exact test two-sided. **d** Scatter dot plot representing the distribution of the length of ssDNA at fork branching points of the indicated strains. 'n' is the total number of DNA replication intermediates analyzed. Values derive from the analysis of the samples of **c**. The middle line in the scatter dot plot indicates the average length of the ssDNA gaps at the fork branching points; the bar shows the distribution range. P-values were calculated by unpaired two-tailed Student's *t* test.

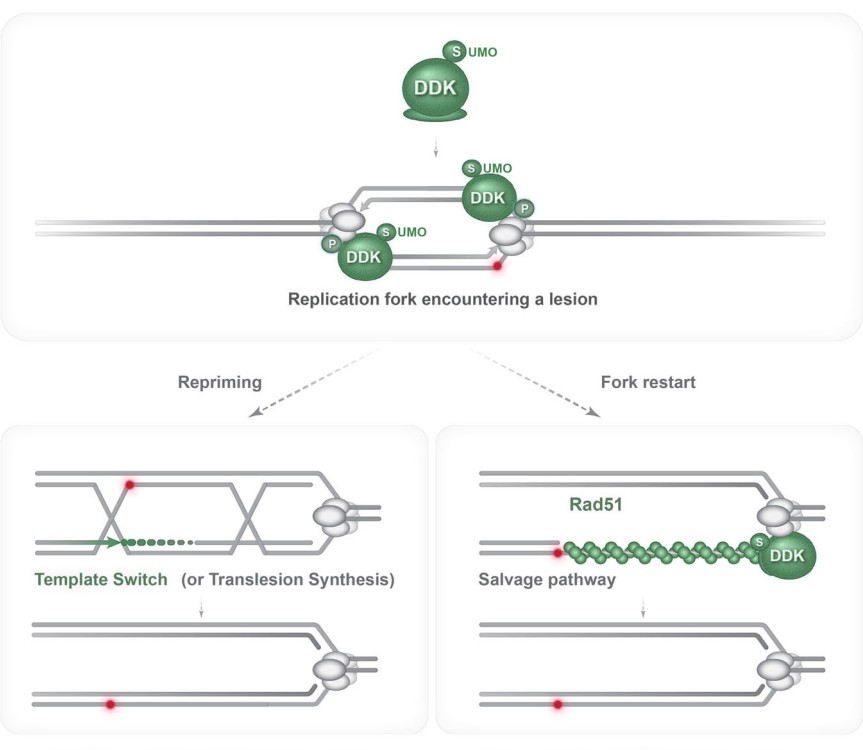

**Fig. 7 Model illustrating SUMOylated DDK kinase as regulator of gap-filling mediated DNA damage tolerance and fork restart by recombination.** The model illustrates monoSUMOylated DDK kinase traveling with replication forks exposed to genotoxic stress. SUMOylated DDK kinase prevents fork uncoupling and stabilizes Rad51 filaments on exposed ssDNA gaps. These filaments may facilitate repriming creating additional substrates for gap-filling in addition to the ones generated on the lagging strand by physiological repriming events. Alternatively, Rad51 assembly on ssDNA stretches at stalled replication fork can be engaged by salvage recombination pathways with potential to generate genome rearrangements. MonoSUMOylated DDK kinase facilitates both recombination-mediated gap-filling and DNA damage tolerance/fork protection in the context of the stalled replication fork.

the indicated timepoints in the presence of 1% sodium azide and incubated in ice for 30 min. Cells were pelleted and washed with ice-cold 1X TE buffer. Genomic DNA was extracted using the genomic DNA extraction kit (QIAGEN) according to manufacturer's instructions. BrdU immunoprecipitation was carried out as previously described in Bermejo et al.[80]. The BrdU containing DNA is sheared to 200–1000 bp fragments by sonication using the Bandelin UW2070 sonicator (5 sonication cycles, 20% power, 15 s/pulse). After each sonication cycle, the chromatin was pelleted by centrifuging at 2300 g for 1 min at 4 °C. Towards the end of sonication, sheared DNA was centrifuged for 5 min at 3000 rpm at 4 °C. The sheared genomic DNA was denatured at 100 °C for 10 min and immediately plunged into ice and were later supplemented with 100 μL of ice-cold PBS 2x and 200 μL of ice cold-PBS with 2% 1x BSA and 0.2% Tween20. BrdU immunoprecipitation was carried out overnight by adding the genomic DNA into 10 μL of Protein A-magnetic Dynabeads (Thermofisher) and 4 μg of anti-BrdU antibody (clone 2B1, MBL, MI-11-3). 10 ml of DNA solution was kept as input. Next day, after immuno-precipitation, beads washed two times with ice-cold lysis buffer (50 mM HEPES-KOH pH7.5, 140 mM NaCl, 1 mM EDTA, 1% Triton x100, 0.1% Na-deoxycholate), two times with ice-cold

lysis buffer + 500 mM NaCl, 2 times with ice-cold washing buffer (10 mM Tris-HCl pH8.0, 250 mM LiCl, 0.5% NP40, 0.5% Na-deoxycholate, 1 mM EDTA) and once with ice-cold 1X TE. The beads are then resuspended in 50 μl of elution buffer (50 mM Tris-HCl pH8, 10 mM EDTA, 1% SDS); and then incubated at 65 °C for 10 min by shaking. After centrifugation and magnetic attachment, the supernatant was transferred into new tubes. In all, 49 μl of 1X TE buffer and 1 ml of Proteinase K (stock 50 mg/mL) was added to both IP and input tubes. The tubes were then mixed without vortexing and were incubated at least 2 h at 37 °C. The DNA from this mixture was purified by PCR purification Kit (QIAGEN), eluted with 50 μL of elution buffer and precipitated at −20 °C overnight in the presence of 40 mM Na-acetate, 1 μL of glycogen and 132.5 μL of 100% ethanol. After centrifugation, the pellet was washed with 70% ethanol and is then re-suspended in 10 μl of double distilled $H_2O$. Input DNA was diluted 50 times and IP DNA 10 times and were proceeded for qPCR as mentioned in the ChIP-qPCR protocol.

**2D gel electrophoresis.** Exponential cells were arrested in G1 phase using alpha factor and synchronously released from G1 phase in 0.033% MMS containing YPD

medium at 28 °C. Samples were collected at the indicated time points and incubated with sodium azide 1% for 30 min on ice. Cells were briefly washed twice with cold water and in vivo psoralen crosslinking and DNA extraction with CTAB were performed as in Giannattasio et al.[9]. The collected cells were resuspended in 5 ml of cold water and transferred to a six-well plate (1 sample/well). In all, 300 μL of 4,5, 8-trimethylpsoralen solution (0.2 mg/ml in EtOH 100%) was added prior to extensive resuspension by pipetting, followed by 5 min of incubation in the dark and then 10 min of UV irradiation at 365 nm (Stratagene UV Stratalinker 2400). The procedure was repeated three more times to ensure extensive crosslinking. Cells were then harvested by centrifugation, washed in cold water, and incubated in spheroplasting buffer (1 M sorbitol, 100 mM EDTA, 0.1% β-mercaptoethanol, and 50 U zymolyase/ml) for 1 h at 30 °C. The spheroplasts are collected by centrifugation and washed once with 10 ml of water. The spheroplast pellets were then resuspended in 2 mL water, 200 mL RNase A (10 mg/ml), and 2.5 mL Solution I (2% w/v cetyltrimethylammonium bromide (CTAB), 1.4 M NaCl, 25 mM EDTA, 100 mM Tris–HCl, pH 7.6) and incubated at 50 °C for 30 min. In all, 200 µl of 20 mg/ml Proteinase K was added to this mixture and the incubation was prolonged at 50 °C again for ~1 h 30 min. After 1.5 h, another 100 µl of Proteinase K was added to this mix and were incubated overnight at 30 °C. Next day, the samples were centrifuged at 4000 rpm for 15 min: the cellular debris pellet was kept for further extraction, while the supernatant was extracted with 2.5 mL chloroform/ isoamylalcohol (24:1) and the DNA in the upper phase was precipitated by addition of 2 volumes of Solution II (1% w/v CTAB, 10 mM EDTA, 50 mM Tris–HCl, pH 7.6) and centrifugation at 9000 rpm for 10 min. The pellet was resuspended in 2 mL Solution III (1.4 M NaCl, 1 mM EDTA, 10 mM Tris–HCl, pH 7.6). Residual DNA from the cellular debris that was kept previously was also extracted by resuspension in 2 ml Solution III and incubation at 50 °C for 30 min, followed by extraction with 1 mL chloroform/isoamylalcohol (24:1). The upper phase was pooled together with the main DNA prep. Total DNA was then precipitated with 1 volume of isopropanol, washed with 70% ethanol, air-dried, and finally resuspended in 250 µl of 10 mM Tris–HCl pH 8 and allowed to dissolve overnight at 4 °C. Genomic DNA extracts were stored at 4 °C. The DNA samples were digested with NcoI, and was precipitated with potassium acetate and isopropanol, and resuspended in 1X TE. The first-dimension gel (500 ml; 0.35% w/v Agarose D1-LE) was prepared with 1x TBE and the digested DNA were run on this gel at 50 V for 18 h at room temperature. The second-dimension gel (500 ml; 0.9% w/v Agarose D1-LE) prepared with 1x TBE was run in the same electrophoresis chamber at 180 V for 8hrs at 4 °C with current limited to 140 mA. DNA molecules separated on the second-dimension gels were then transferred onto nylon filters via Southern blotting following standard procedures.

The signals were detected following 2D gel electrophoresis and standard southern blot procedures using probes against and ARS305 (Chr. III 39,026–41,647) radiolabeled according to the protocol of the Prime-A-Gene labeling system and purified with Probe Quant G-50 micro columns. The images were acquired using Amersham Typhoon Scanner software V1.0 and images were prepared using the ImageJ software. Quantification of 2D gels was performed as described in Fumasoni et al.[6].

**Electron microscopy.** Yeast cell sample collection, in vivo psoralen crosslinking and DNA extraction was performed based on the CTAB Psoralen procedure described in 2D gel electrophoresis protocol. The replication intermediates (RIs) were enriched as previously described[44,45]. Briefly, 15 μg of extracted genomic DNA was subjected to restriction digestion using *PvuI* (NEB) following manufacturer's instructions, then further treated with RNAse III to avoid double-strand RNA contamination of the samples (single-strand RNA was removed by RNase A treatment during the genomic DNA extraction). This mix was then adjusted to 300 mM NaCl and loaded on a chromatography column containing 1 ml of BND cellulose stock (0.1 g/column; Sigma B-6385, pre-equilibrated with 10 mM Tris–HCl pH 8, 300 mM NaCl) to enrich the DNA fragments containing ssDNA. BND cellulose and DNA were incubated together for 30 min with resuspension every 10 min to allow full binding of the DNA molecules and the flowthrough obtained was collected by gravity flow. In all, 1 ml of 10 mM Tris–HCl pH 8 containing 1 M NaCl was added to the column to collect linear double-stranded molecules (salt elution). This step was repeated twice to ensure 70–90% elution of linear DNA. The DNA enriched for ssDNA-containing RIs were eluted from the column by the addition of 600 µl of 10 mM Tris–HCl pH 8, 1 M NaCl containing 1.8% caffeine, followed by 10 min of incubation time. Caffeine fractions were purified, to change the buffer and to reduce the concentration of caffeine in the samples, by using conical Amicon Ultra centrifugal filters (0.5 ml 100K-MWCO 100 K). The RIs purified through the Amicon filters were concentrated in 25–30 microliters of 10 mM Tris–HCl pH 8. Aliquotes of caffeine fractions carrying DNA samples enriched for RIs were spread onto a water surface in a mono-molecular layer in the presence of benzyl-dimethyl-alkyl-ammonium chloride (BAC) and the DNA molecules in the monomolecular layer were adsorbed on carbon-coated metal grids (4-nm thickness). The adsorbed DNA fibers on the carbon surface were subsequently stained with uranyl acetate dissolved in ethanol. The DNA molecules stained with uranyl acetate were subsequently subjected to low-angle rotary shadowing with 8 nm of platinum as previously described[44,45]. For the low angle rotary shadowing we utilized a Leica MED020 e-beam evaporator equipped with oil-free deep vacuum system (membrane and turbo-molecular pumps), the EVM030 control unit with two EK030 electron beam evaporation sources (Leica catalogue number 16BU007086-T),

the QSG100 film thickness monitor (Leica 16LZ03428VN), the QSK060 Quartz Head (Leica 16LZ03440VN), the Tiltable Rotary Stage (Leica 16BU007283T), and the high precision rotation plate PR01 (ThorLabs Newton, New Jersey, USA). The EM images presented in this study were acquired using an FEI Tecnai 12 G2 Bio-twin electron microscope (run at 120KV) with a side-mounted optical-fibered GATAN ORIUS SC-1000 camera (11 mega-pixels). The camera was operated in the stand-alone configuration by the GATAN Digital Micrograph Suite Software (Gatan, version 2.3.3 64 bit) and the montage plug-in (Gatan, 64-bit version). EM pictures raw files were saved in dm3 format. The ImageJ (version 2.1.0/1.53 C) software (open source) was utilized to analyze the raw EM pictures. The average thickness of the DNA fiber in this specific experimental condition was distributed around 10 nm. The conversion factor for the calculation of the DNA fiber length was 0.36 nm/base pair and was established through the measurement of the length of plasmid DNA molecules of known size utilized as internal standards. The pixel size was calibrated at each magnification using the GATAN digital micrograph software. The experimental conditions utilized in this study allow the distinction between dsDNA and ssDNA based on the different thickness of the DNA fiber[9].

We note that the length of the ssDNA stretches, calculated using the same conversion factors for both dsDNA and ssDNA, might have been underestimated by a maximum experimentally determined factor of 10–15%. This factor was calculated for extreme conditions of long and entirely single-stranded DNA molecules that show reduced anchorage to the carbon layer and have different stretching properties.

**Reporting summary.** Further information on research design is available in the Nature Research Reporting Summary linked to this article.

## Data availability
The data generated by this study are provided in the main text and figures and the accompanying Supplementary Information and are available from the corresponding author upon reasonable request. Source data are provided with this paper and in Mendeley Data (DOI: 10.17632/c284kmxv6x.1).

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

## Acknowledgements

We thank Felix Prado and G. Liberi for sharing strains, Branzei team members for sharing unpublished reagents, demonstrating techniques and critical discussions and to Barnabas Szakal for additional help with the artwork. We thank the Imaging facility and the EM TDU/DNA single molecules unit of IFOM for technical support. This study was supported by the Italian Association for Cancer Research (AIRC IG 18976 and IG 23710) and European Research Council (Consolidator Grant 682190) grants to D.B. C.R.J. was partly supported by the AIRC 23998 fellowship.

## Author contributions

D.B. conceived the study. D.B. and C.R.J. designed the experiments. C.R.J. performed the experiments. S.D. and M.G. acquired and analyzed the data for EM experiments. C.R.J., S.D., M.G., and D.B. constructed the figures. D.B. wrote the paper with inputs from all co-authors.

## Competing interests

The authors declare no competing interests.
