## [Peer Review File · Nature Communications]

REVIEWER COMMENTS

Reviewer #1 (Remarks to the Author):

DNA damage tolerance (DDT) can proceed by translesion synthesis or homologous recombination through template switching. Much of what is known about these processes and the relationships between the two major pathways comes from studies using yeast, and in particular the Branzei lab, which has delineated the HR template switch pathway and the role of HR proteins in this pathway. This lab has also identified a salvage pathway that is HR-dependent but differs from the template switch HR pathway in that PCNA modification by ubiquitylation is not needed. Here the role of DDK kinase in the HR template switch versus the HR salvage pathway is presented. The authors show that DDK is needed for recombination-mediated DDT and that this requires multi-monoSUMOylation. In the absence of functional template switching due to defects in DDK kinase activity, monoSUMOylation, and SIM mutations, the replication fork becomes uncoupled and leaves large (visible by EM) single strand regions at the replication fork junction. The salvage pathway can rescue the uncoupled fork. The authors conclude that SUMOylated DDK acts with the Rad51 recombinase and its SIMs to prevent replication fork uncoupling and promote HR mediated gap-filling.

These are complex but important conclusions that add new dimensions to our understanding of DDT. Although the experiments reported here are done in yeast, because the genetics and ability to assess replication fork structures by 2D gels lends itself to a detailed analysis of the problem, the findings are universal in that the pathways and proteins are conserved. Moreover, DDT is essential for tolerance of replication stress and exogenous DNA damage in cells and defects are associated with cancers, particularly skin cancers, in humans.

Figure 1 present two methods of reducing DDK/Cdc7 activity, though use of a ts mutant at semi-permissive temperature and use of an inhibitor of Cdc7 kinase activity. Both show reduction in X structures in the absence of Sgs1, but neither condition completely reduced X structures/SCJ. Is this because the Cdc7 activity is not fully depleted, or because there is another pathway that gives the same X-shaped DNA molecule? The authors are cautious and say that the DDK facilitates template switching.

Figure 2 shows a clever way using restriction enzyme site heterozygosity in diploids to distinguish between sister versus interhomolog junctions. It would be helpful to indicate on one real gel image where the red arrow interhomolog junction would migrate. I think the importance of this experiment is sort of lost using yeast as most of the experiments here are done in haploids, but for mammalian cells, they are diploid and therefore can theoretically form many interhomolog junctions. This point could be mentioned in the setup of this experiment on pages 6-7.

Figure 3 shows micrographs of DNA spreads and has a scale bar. Under the spreading conditions used is this the same for double strand DNA and single strand DNA? Just nit-picking here if the length of ssDNA at the fork is important (Figure 3c and e). Obviously, there is a difference between the different strains.

Figure 5 examines the salvage pathway. Here loss of activation of the salvage pathway through SIZ1 deletion gives about the same level of X molecules as reduction of DDK activity (Figure 5a). Again, where

are the residual X molecules coming from? Do stalled forks give X molecules too?

Figure 6 examines the role of the Rad51 SIMs in mediating HR damage bypass and preventing fork uncoupling. Here the X molecules are virtually eliminated in the rad51 mutants. Therefore, in the previous figures are X molecules being formed at a low rate through the normal pathways, or is there another Rad51-mediated pathway?

If I understand the arguments and Figure 7 correctly, template switching or translesion synthesis fills in gaps that are left behind the replication fork. These gaps do not arise from fork uncoupling. Additionally, they have a 5' and a 3' end (gap) that renders them easy to repair/fix in via an error-prone mode (TLS) or an error-free mode (HR-dependent template switch). The salvage pathway acts at gaps at the fork that arise from fork uncoupling and have only one ss end, which makes them hard to repair and can lead to a DSB formation through nicking of the template strand which is now in a ssDNA configuration. This is why the salvage pathway has to be carefully controlled. If this interpretation is correct, the authors could add a few lines in the discussion on the differences between the HR-mediated DDT pathways and substrates.

This is a very thoughtful paper that takes some concentration to get through. It is relevant to mammalian cells and human disease.

Reviewer #2 (Remarks to the Author):

The manuscript entitled „Rad51-mediated replication of damaged templates relies on monoSUMOylated DDK kinase“ by Joseph et al., characterizes functions of DDK kinase in replication-associated processes induced by stress. By using CDC7 alleles which abolish or inhibit the kinase activity they show its role in facilitating recombination-mediated repair. Similar role they also observed for monoSUMOylated DDK, potentially by attracting SUMO-interaction motif containing factors required for this process. Replication-associated recombination does not seem to affect the partner choice with strong bias towards sister chromatid. Both DDK properties are also required to prevent replication fork uncoupling as indicated by increase gapped forks in DDK mutants rather than nuclease-mediated resection of the replication forks. The characterization of the mechanism by which DDK promotes recombination-mediated DDT reveals role in Rad18- and Mms2-dependent gap filling through template switching mediated by Rad51. The authors show that DDK activity and monoSUMOylation also promotes a parallel recombination salvage pathway, by facilitating recruitment and retention of Rad51 proximal to replication origins. The manuscript also provides novelty about the context in which this salvage pathway occurs, suggesting stalled replication forks rather than gaps left behind the replication as preferred place. Finally, the authors uncover that previously identified Rad51 SUMO interaction motifs play a major role in mediating recombination-dependent damage bypass and in preventing uncoupling replication forks upon stress.

This is very well structured and written manuscript with clearly defined hypothesis and corresponding experiments delineating role of DDK in replication stress associated processes. I have two comments/suggestions for the authors:

1. Can authors show that SIM mutants of Rad51 are otherwise functional, either genetically in DSB repair or biochemically in D-loop/strand exchange? This is needed to rule out possible loss of function, given very strong phenotypes similar rather to rad51 deletion than siz1 deletion (Figure 5. and 6).

2. Srs2 as a major antirecombinase is also known to be SUMOylated and interacts with SUMO, PCNA and Rad51. Would Srs2 mutants defective in these interactions or SUMOylation help clarify the role of Srs2 in regulation of Rad51-mediated replication of damaged templates by alleviating/promoting the DDK phenotypes?

Reviewer #3 (Remarks to the Author):

Review of Joseph et al. "Rad51-mediated replication of damaged templates relies on monoSUMOylated DDK kinase" (NCOMMS-22-02752)

In this interesting work, the Branzei lab has investigated functions of Cdc7/Dbf4 (DDK) in DNA replication-associated processes triggered by the presence of DNA lesions. Through a combination of genetic and molecular approaches, they convincingly show that DDK has an important novel role in recombination-mediated DNA damage tolerance. They show that this DDK function requires kinase activity and its mono-SUMOylation and helps prevent fork uncoupling. Their data indicate that mono-SUMOylated DDK promotes gap-filling mediated by Rad51, for which the SUMO interacting motifs of this protein are required. Of note, they also uncover that the salvage pathway of recombination operates at damaged stalled forks to alleviate accumulation of ssDNA. The experimental work is well done and the conclusions are sound. The paper addresses an important issue that should be of broad interest to those working on the mechanisms of genome integrity maintenance during cell proliferation. I am therefore in clear support of publication of this work in Nat. Commun.

Some comments/suggestions that the authors might take in consideration:

- Page 3, lines 63-65. 'However, how the two recombination-dependent DDT pathways are regulated and ... are not yet understood'. I would say something like '...are only partially understood', as some progress has been made in recent years.

- I understand that 28°C is used in S-phase throughout the paper because it is a semi-permissive temperature for the conditional cdc7-4 mutant that allows cell proliferation but still induces DDK-dependent defects. I think it might help the general reader, especially those not familiar with yeasts, to clearly explain this approach at the beginning of the Results section.

- Fig. 2a. Correct the labelling of the immunoblots. Cells were not treated with MMS+Tc in G1.

- Fig. 2b. For consistency with the rest of the figures, I think it would be convenient to show quantification of the experiment if possible.

- Nucleases experiments to analyse possible resection. I would describe the mre11-H125N mutant used - I understand its nuclease activity is inactivated but not its other repair functions and that is why it was preferred over a null mutant.

- Page 8, lines 176-177. 'Of interest, these gapped forks do not impair proliferation'. I do not completely understand. I suppose that treatment with 0.033% MMS, which as the authors show causes increased number of gapped forks in DDK mutants, allows slow S-phase progression but not cell proliferation (one might expect that cells stop in G2/M under these DNA-damaging conditions). Do they mean gapped forks do not impair bulk DNA replication?

- Fig. 4a and Supplementary Fig. 3a,b. It is interesting that the combination of *cdc7-4* with deletions of RAD18 or MMS2 does not aggravate the defect in SCJ formation seen in *cdc7-4* cells, while with regard to MMS sensitivity there are additive effects. Can the authors discuss these observations a little more?

- Page 9, lines 202-204, Fig. 4c and Supplementary Fig. 3c. From the drop-assay images shown, it is clear that *ddk-KR* suppresses the MMS sensitivity of *mms2Δ* cells (Fig. 4C), but it is not that obvious for *rad18Δ*. If what *ddk-KR* rescues is the consequences of the absence of PCNA-polyubiquitylation, then the *ddk-KR mms2Δ* result seems to be sufficient anyway.

- Fig. 5b (left). ChIP-qPCR. Is *rad51Δ* a background control?

- *rad51* SIM mutants. Do the authors know if the mutant proteins levels are similar to those of the wt so that the result obtained can be confidently attributed to the mutations?

- While reading the paper, one could think of the possibility of a direct interaction between SUMO-DDK and Rad51 through the SIM motifs, but the authors say that this is not the case, suggesting a possible Mcms-mediated interaction. Could they elaborate further in the Discussion on this idea and its potential implications to prevent fork uncoupling? Do they consider other potential interaction scenarios, involving other proteins?

First, we like to thank the reviewers for the careful evaluation of our work. We are very happy that all the reviewers appreciated the depth and impact of our findings and for the suggestions to improve the manuscript. We addressed the suggestions that were raised as detailed in the point-by-point response below. We hope that the reviewers will now be happy to recommend our revised manuscript for publication.

REVIEWER COMMENTS

Reviewer #1 (Remarks to the Author):

DNA damage tolerance (DDT) can proceed by translesion synthesis or homologous recombination through template switching. Much of what is known about these processes and the relationships between the two major pathways comes from studies using yeast, and in particular the Branzei lab, which has delineated the HR template switch pathway and the role of HR proteins in this pathway. This lab has also identified a salvage pathway that is HR-dependent but differs from the template switch HR pathway in that PCNA modification by ubiquitylation is not needed. Here the role of DDK kinase in the HR template switch versus the HR salvage pathway is presented. The authors show that DDK is needed for recombination-mediated DDT and that this requires multi-monoSUMOylation. In the absence of functional template switching due to defects in DDK kinase activity, monoSUMOylation, and SIM mutations, the replication fork becomes uncoupled and leaves large (visible by EM) single strand regions at the replication fork junction. The salvage pathway can rescue the uncoupled fork. The authors conclude that SUMOylated DDK acts with the Rad51 recombinase and its SIMs to prevent replication fork uncoupling and promote HR mediated gap-filling.

These are complex but important conclusions that add new dimensions to our understanding of DDT. Although the experiments reported here are done in yeast, because the genetics and ability to assess replication fork structures by 2D gels lends itself to a detailed analysis of the problem, the findings are universal in that the pathways and proteins are conserved. Moreover, DDT is essential for tolerance of replication stress and exogenous DNA damage in cells and defects are associated with cancers, particularly skin cancers, in humans. We are happy that the reviewer finds that our study reports important conclusions for understanding DNA damage tolerance and its regulation.

Figure 1 present two methods of reducing DDK/Cdc7 activity, though use of a *ts* mutant at semi-permissive temperature and use of an inhibitor of Cdc7 kinase activity. Both show reduction in X structures in the absence of Sgs1, but neither condition completely reduced X structures/SCJ. Is this because the Cdc7 activity is not fully depleted, or because there is another pathway that gives the same X-shaped DNA molecule? The authors are cautious and say that the DDK facilitates template switching.

The first condition only partly perturbs Cdc7 function, as the temperature used is fully permissive for growth and cell cycle proliferation (Fig. 1b, FACS profiles) but still causes DNA repair defects (Supplementary Fig. 3b). The second condition based on *cdc7-as3* inactivates the kinase at a point in which replication initiation and progression through S phase are not visibly affected but at which the recombination process has already initiated. Therefore, for the part of the experiment without inhibitor, X-molecules formed normally. Because of these features, we can only expect a partial effect of Cdc7 perturbations on the X molecules. We revised the text to ensure that these points are accurately described. It is also important to note

that a regulator of the recombination process, differently from a core factor such as Rad51, would most likely facilitate the process but not completely abolish it unless it completely inactivates one core activity. Because of the mild nature of the Cdc7 mutations employed, this however, cannot be the case. We also like to note that while inactivation of core recombination factors is frequently not compatible with cellular proliferation, dysfunctions in regulators of the recombination process is often detected in tumor cells.

Figure 2 shows a clever way using restriction enzyme site heterozygosity in diploids to distinguish between sister versus interhomolog junctions. It would be helpful to indicate on one real gel image where the red arrow interhomolog junction would migrate. I think the importance of this experiment is sort of lost using yeast as most of the experiments here are done in haploids, but for mammalian cells, they are diploid and therefore can theoretically form many interhomolog junctions. This point could be mentioned in the setup of this experiment on pages 6-7.

We thank the reviewer for this suggestion, which we now incorporated in the text. As suggested by the reviewer, we are also indicating with arrows on real gels where the interhomolog junctions migrate.

Figure 3 shows micrographs of DNA spreads and has a scale bar. Under the spreading conditions used is this the same for double strand DNA and single strand DNA? Just nit-picking here if the length of ssDNA at the fork is important (Figure 3c and e). Obviously, there is a difference between the different strains.

The scale bar is indeed calculated for dsDNA, using dsDNA molecules of known length as internal standards. The point raised by the reviewer is well taken, it was our omission to mention it. We now reported in the figure legend that scale bars are obtained for dsDNA. Nevertheless, as the reviewer acknowledges, this is a good approximation also for ssDNA length, with clear differences between the different strains. The exact discrepancy between the conversion factor calculated for dsDNA and ssDNA was reported in Giannattasio et al, Nat Struct Mol Biol, 2014 and in references therein. Now we discussed these points as well in the Methods section of the manuscript. Specifically, we report that the length of the ssDNA stretches, calculated using the same conversion factors for both dsDNA and ssDNA, might have been underestimated by a maximum experimentally determined factor of 10-15%. This was calculated for extreme conditions of long and entirely single-stranded DNA molecules (ssDNA genomes of phages) that show reduced anchorage to the carbon layer and have different stretching properties. In physiological conditions, ssDNA stretches present on DNA replication, recombination and repair intermediates are never so long and always surrounded by dsDNA regions.

Figure 5 examines the salvage pathway. Here loss of activation of the salvage pathway through SIZ1 deletion gives about the same level of X molecules as reduction of DDK activity (Figure 5a). Again, where are the residual X molecules coming from? Do stalled forks give X molecules too?

Here, DDK mutations reduce the X molecules to the same degree, regardless of the *SIZ1* status. The *SIZ1* deletion reduces PCNA SUMOylation, inhibiting the recruitment of the Srs2 anti-recombinase and thus allowing “salvage” recombination. The levels of X-molecules in *sgs1* and *sgs1 siz1* backgrounds are very similar to each other because the accumulation is primarily dictated by failure to resolve these intermediates, forming via different pathways, via Sgs1-Top3. DDK mutations reduce the levels of X-molecules in both *sgs1* and *sgs1 siz1* backgrounds (Fig. 5a). We cannot assign a specific class of X molecules in 2D gels to arise at stalled forks. The reduction in PCNA SUMOylation (now further extended to loss of PCNA) generates

recombination through the salvage pathway. While the structural features of the X molecules arising primarily postreplicatively from the error-free branch of DNA damage tolerance dependent on PCNA polyubiquitination are known and they result to be composed of pseudo-double Holliday Junctions (Wong et al, Mol Cell, 2020; Giannattasio et al, NSMB, 2014), the structure of the X-molecule generated from the salvage pathway in the absence of PCNA-SUMOylation is still under investigation. Error-prone recombination events will likely happen at stalled forks too, but they will be too rare to be detected as a specific signal in 2D gel. Regarding the contribution of DDK mutants, the effect is partial because the employed DDK mutations are hypomorphic (see also Fig. 1) and they only diminish the efficiency and stability of the Rad51 filament (see also Fig. 5b).

Figure 6 examines the role of the Rad51 SIMs in mediating HR damage bypass and preventing fork uncoupling. Here the X molecules are virtually eliminated in the rad51 mutants. Therefore, in the previous figures are X molecules being formed at a low rate through the normal pathways, or is there another Rad51-mediated pathway?

Rad51 is a core factor of the error-free DNA damage tolerance pathway. So far, all the X-molecules generated during the different branches of DNA damage bypass are fully dependent on Rad51 for their formation. Figure 6 highlights that the motifs encoding the putative SIMs of Rad51 are also critical for Rad51 function in X molecule formation but more analysis and biochemical characterization will be needed in the future to understand their functionality. In the previous figures, the X-molecules are formed less efficiently because Rad51 function and recruitment is only partially affected by the employed Cdc7 mutations (see Fig. 5b), as opposed to rad51-SIM mutations that severely reduce Rad51 recruitment (see Supplementary Fig. 4d). They also cause drastic MMS sensitivity (in contrast to mild or lack of sensitivity of the DDK mutants).

If I understand the arguments and Figure 7 correctly, template switching or translesion synthesis fills in gaps that are left behind the replication fork. These gaps do not arise from fork uncoupling. Additionally, they have a 5' and a 3' end (gap) that renders them easy to repair/fix in via an error-prone mode (TLS) or an error-free mode (HR-dependent template switch). The salvage pathway acts at gaps at the fork that arise from fork uncoupling and have only one ss end, which makes them hard to repair and can lead to a DSB formation through nicking of the template strand which is now in a ssDNA configuration. This is why the salvage pathway has to be carefully controlled. If this interpretation is correct, the authors could add a few lines in the discussion on the differences between the HR-mediated DDT pathways and substrates.

Yes, this interpretation is largely correct and we expanded the discussion in pointing out the fragility of ssDNA and the DSB pathways operating to repair collapsed forks. We also note that persistent ssDNA gaps at the fork junctions may facilitate erroneous recombination reactions primed by annealing of the ssDNA at the fork junction with other genomic regions, or through formation of reversed forks with exposed 3' ends ready to invade other genomic regions, potentially causing error-prone outcomes.

This is a very thoughtful paper that takes some concentration to get through. It is relevant to mammalian cells and human disease.

We thank the reviewer for finding our paper thoughtful and relevant also to other systems and for human disease.

Reviewer #2 (Remarks to the Author):

The manuscript entitled „Rad51-mediated replication of damaged templates relies on monoSUMOylated DDK kinase“ by Joseph et al., characterizes functions of DDK kinase in replication-associated processes induced by stress. By using CDC7 alleles which abolish or inhibit the kinase activity they show its role in facilitating recombination-mediated repair. Similar role they also observed for monoSUMOylated DDK, potentially by attracting SUMO-interaction motif containing factors required for this process. Replication-associated recombination does not seem to affect the partner choice with strong bias towards sister chromatid. Both DDK properties are also required to prevent replication fork uncoupling as indicated by increase gapped forks in DDK mutants rather than nuclease-mediated resection of the replication forks. The characterization of the mechanism by which DDK promotes recombination-mediated DDT reveals role in Rad18- and Mms2-dependent gap filling through template switching mediated by Rad51. The authors show that DDK activity and monoSUMOylation also promotes a parallel recombination salvage pathway, by facilitating recruitment and retention of Rad51 proximal to replication origins. The manuscript also provides novelty about the context in which this salvage pathway occurs, suggesting stalled replication forks rather than gaps left behind the replication as preferred place. Finally, the authors uncover that previously identified Rad51 SUMO interaction motifs play a major role in mediating recombination-dependent damage bypass and in preventing uncoupling replication forks upon stress.

This is very well structured and written manuscript with clearly defined hypothesis and corresponding experiments delineating role of DDK in replication stress associated processes. I have two comments/suggesting for the authors:

We are happy that the reviewer finds our work clear and impactful and for the suggestions to improve the manuscript and discussion.

1. Can authors show that SIM mutants of Rad51 are otherwise functional, either genetically in DSB repair or biochemically in D-loop/strand exchange? This is needed to rule out possible loss of function, given very strong phenotypes similar rather to rad51 deletion than siz1 deletion (Figure 5. and 6).

The potential loss of function of the SIM mutants was a point of potential concern for us as well as initially we established the putative SIM mutants based on their loss of interaction with SUMO. Now we identified that the SIM mutants have only mild sensitivity in response to zeocin, which causes DSB formation, although they are nearly loss of function for MMS-induced replication lesions (Fig. 6a). The mutants have normal protein stability (Supplementary Fig. 5b). Moreover, in line with studies in mammalian cells, we see very reduced recruitment of these Rad51 mutants to replication sites upon MMS treatment (Supplementary Fig. 5c), likely because of their defective interaction with Rad52, as reported by Bergink et al, Nat Cell Biol, 2013. We were not aware of this latter paper, in which the authors characterize the rad51-SIM mutant corresponding to SIM2, showing to have reduced interaction with Rad52, but normal interaction with Rad54. Moreover, the authors show that rad51-SIM2 mutant can be suppressed by loss of Rad52 SUMOylation, interpreting this result as to that SUMOylated Rad52 may form toxic SUMO-SIM interactions if the SIM receptor in Rad51 is not present. We extended the discussion in this sense and added the results mentioned above. Moreover, we are careful and refer to these mutants as “putative SIM mutants” and indicate the requirement of their further biochemical characterization in future studies.

2. Srs2 as a major antirecombinase is also known to be SUMOylated and interacts with SUMO, PCNA and Rad51. Would Srs2 mutants defective in these interactions or SUMOylation help clarify the role of Srs2 in regulation of Rad51-mediated replication of damaged templates by alleviating/promoting the DDK phenotypes?

Srs2 is indeed a critical player in recombination affecting also other aspects of DSB repair and checkpoint signaling. The genetics and biochemistry of Srs2 is complex, making it hard to assign a specific interaction and mutation described so far to a single process. There is also a very important technical shortcoming in extending the current analysis as we do not have a straightforward way to screen for Srs2 functions in regulating salvage recombination, except for the very laborious EM analysis. We added during the revision the result that Srs2 loss phenocopies *siz1Δ* (Supplementary Fig. 4a, b). The comment of the reviewer is insightful and it is an aspect that is indeed worth considering. Moreover, because Srs2 itself may be modulated by DDK (along or jointly with CDK, Saponaro et al, PLoS Genetics, 2010), the effect of DDK on Rad51 recruitment may be indirect, via Srs2. This is another aspect we are commenting now in the discussion.

Reviewer #3 (Remarks to the Author):

Review of Joseph et al. "Rad51-mediated replication of damaged templates relies on monoSUMOylated DDK kinase" (NCOMMS-22-02752)

In this interesting work, the Branzei lab has investigated functions of Cdc7/Dbf4 (DDK) in DNA replication-associated processes triggered by the presence of DNA lesions. Through a combination of genetic and molecular approaches, they convincingly show that DDK has an important novel role in recombination-mediated DNA damage tolerance. They show that this DDK function requires kinase activity and its mono-SUMOylation and helps prevent fork uncoupling. Their data indicate that mono-SUMOylated DDK promotes gap-filling mediated by Rad51, for which the SUMO interacting motifs of this protein are required. Of note, they also uncover that the salvage pathway of recombination operates at damaged stalled forks to alleviate accumulation of ssDNA. The experimental work is well done and the conclusions are sound. The paper addresses an important issue that should be of broad interest to those working on the mechanisms of genome integrity maintenance during cell proliferation. I am therefore

in clear support of publication of this work in Nat. Commun.

We are very happy that the reviewer finds our work and conclusions important and of broad interest, supporting its publication in *Nature Communications*.

Some comments/suggestions that the authors might take in consideration:

- Page 3, lines 63-65. 'However, how the two recombination-dependent DDT pathways are regulated and ... are not yet understood'. I would say something like '...are only partially understood', as some progress has been made in recent years.

We corrected this.

- I understand that 28°C is used in S-phase throughout the paper because it is a semi-permissive temperature for the conditional *cdc7-4* mutant that allows cell proliferation but still induces DDK-dependent defects. I think it might help the general reader, especially those not familiar with yeasts, to clearly explain this approach at the beginning of the Results section.

We have added comments in this regard.

- Fig. 2a. Correct the labelling of the immunoblots. Cells were not treated with MMS+Tc in G1.

We corrected the labeling of the Western blots.

- Fig. 2b. For consistency with the rest of the figures, I think it would be convenient to show quantification of the experiment if possible.

We added quantification of the IHJs.

- Nucleases experiments to analyse possible resection. I would describe the *mre11-H125N* mutant used -I understand its nuclease activity is inactivated but not its other repair functions and that is why it was preferred over a null mutant.

Indeed, this is correct. We do not know if other repair functions are not affected, but it is less pleiotropic than *mre11* delta.

- Page 8, lines 176-177. 'Of interest, these gapped forks do not impair proliferation'. I do not completely understand. I suppose that treatment with 0.033% MMS, which as the authors show causes increased number of gapped forks in DDK mutants, allows slow S-phase progression but not cell proliferation (one might expect that cells stop in G2/M under these DNA-damaging conditions). Do they mean gapped forks do not impair bulk DNA replication? The gapped forks do not impair bulk DNA replication, and they may not affect proliferation either as *cdc7-4* and *ddk-KR* mutants are either mildly sensitive or lack sensitivity to MMS. We commented on this aspect and clarified the concept.

- Fig. 4a and Supplementary Fig. 3a,b. It is interesting that the combination of *cdc7-4* with deletions of RAD18 or MMS2 does not aggravate the defect in SCJ formation seen in *cdc7-4* cells, while with regard to MMS sensitivity there are additive effects. Can the authors discuss these observations a little more?

The MMS sensitivity encompasses defects in the recombination and mutagenic modes of DDT. *cdc7-4* was shown to be defective in mutagenic bypass pathways. However, these mutagenic pathways are likely functional upon Rad18 and Mms2 loss resulting in additivity with *cdc7-4* for MMS sensitivity. The SCJ formation measures the defect in recombination-mediated gap-filling, for which no additivity is observed, thus highlighting a role for DDK in this process along Rad18 and Mms2.

- Page 9, lines 202-204, Fig. 4c and Supplementary Fig. 3c. From the drop-assay images shown, it is clear that *ddk-KR* suppresses the MMS sensitivity of *mms2Δ* cells (Fig. 4C), but it is not that obvious for *rad18Δ*. If what *ddk-KR* rescues is the consequences of the absence of PCNA-polyubiquitylation, then the *ddk-KR mms2Δ* result seems to be sufficient anyway. It is true that the suppression effect of *ddk-KR* is much better observed in *mms2Δ*, although a mild suppression is observed for *rad18Δ* as well.

- Fig. 5b (left). ChIP-qPCR. Is *rad51Δ* a background control?

Yes, indeed, we clarified this in the text.

- *rad51* SIM mutants. Do the authors know if the mutant proteins levels are similar to those of the wt so that the result obtained can be confidently attributed to the mutations?

Yes, we added relevant Western blots experiments in Supplementary Fig. 4b.

- While reading the paper, one could think of the possibility of a direct interaction between SUMO-DDK and Rad51 through the SIM motifs, but the authors say that this is not the case, suggesting a possible Mcms-mediated interaction. Could they elaborate further in the Discussion on this idea and its potential implications to prevent fork uncoupling? Do they consider other potential interaction scenarios, involving other proteins?

Indeed, a most straightforward scenario would be that SUMO-DDK interacts with one or both of the Rad51 SIM motifs. However, we do not observe interaction between Rad51 and DDK subunits in co-IP or two hybrid experiments. As MCM was shown to interact with both Rad51 and (SUMOylated)-DDK, we suggest that MCM may bridge the interaction. In addition, another possibility we are now discussing in the manuscript is that SUMOylated DDK may downregulate Srs2 function to allow salvage pathway to proceed effectively. This latter possibility, along other possible DDK targets, will need to be addressed however by other future studies.

REVIEWERS' COMMENTS

Reviewer #1 (Remarks to the Author):

The authors have given an excellent response to the reviewers' comments and have addressed all of the issues. The paper represents an important advance in our understanding on the role of DDK kinase in DNA damage tolerance. The paper highlights the difference between template switching and the salvage recombination pathway and the location of ssDNA gaps and replication fork coupling/uncoupling. These tolerance pathways are often overlooked in mammalian cell studies where at "HR" mechanisms using RAD51 are lumped together. This paper should highlight the different pathways and the benefits/dangers of each.

Reviewer #2 (Remarks to the Author):

Authors have satisfactorily addressed my comments/suggestion and I recommend the manuscript for publication in Nature Communications.

Reviewer #3 (Remarks to the Author):

The authors have nicely addressed all the minor comments I raised after the first submission. In my opinion, they have also satisfactorily discussed all the points raised by the other two reviewers. I believe they have improved what was already an excellent study that significantly helps to understand the complex mechanisms of DNA damage tolerance in eukaryotic cells. I anticipate that this work will have a major impact in the fields of DNA repair/recombination/replication and therefore have no doubt that it should be published in Nat. Commun. in its present form.

We like to thank the reviewers for the careful evaluation of our work. We are very happy that all the reviewers appreciated the depth and impact of our findings and recommended our revised manuscript for publication in *Nature Communications*.

REVIEWER COMMENTS

Reviewer #1 (Remarks to the Author):

The authors have given an excellent response to the reviewers' comments and have addressed all of the issues. The paper represents an important advance in our understanding on the role of DDK kinase in DNA damage tolerance. The paper highlights the difference between template switching and the salvage recombination pathway and the location of ssDNA gaps and replication fork coupling/uncoupling. These tolerance pathways are often overlooked in mammalian cell studies where at "HR" mechanisms using RAD51 are lumped together. This paper should highlight the different pathways and the benefits/dangers of each.

We are happy to read the positive comments of the reviewers on the impact that our study makes on understanding how recombination is regulated during replication.

Reviewer #2 (Remarks to the Author):

Authors have satisfactorily addressed my comments/suggestion and I recommend the manuscript for publication in *Nature Communications*.

We are happy that the reviewer is satisfied by the revision and recommends our paper for publication in *Nature Communications*.

Reviewer #3 (Remarks to the Author):

The authors have nicely addressed all the minor comments I raised after the first submission. In my opinion, they have also satisfactorily discussed all the points raised by the other two reviewers. I believe they have improved what was already an excellent study that significantly helps to understand the complex mechanisms of DNA damage tolerance in eukaryotic cells. I anticipate that this work will have a major impact in the fields of DNA repair/recombination/replication and therefore have no doubt that it should be published in *Nat. Commun.* in its present form.

We are very happy that the reviewer finds our work and conclusions important and of broad interest and recommends its publication in *Nature Communications*.